# Single-layer spatial analog meta-processor for imaging processing

Zhuochao Wang[1,8], Guangwei Hu[2,8], Xinwei Wang[1,8], Xumin Ding[1,3,4,5✉], Kuang Zhang[6✉], Haoyu Li[1✉], Shah Nawaz Burokur[7✉], Qun Wu[6], Jian Liu[1,3,4,5], Jiubin Tan[1,3] & Cheng-Wei Qiu[2✉]

Computational meta-optics brings a twist on the accelerating hardware with the benefits of ultrafast speed, ultra-low power consumption, and parallel information processing in versatile applications. Recent advent of metasurfaces have enabled the full manipulation of electromagnetic waves within subwavelength scales, promising the multifunctional, high-throughput, compact and flat optical processors. In this trend, metasurfaces with nonlocality or multi-layer structures are proposed to perform analog optical computations based on Green's function or Fourier transform, intrinsically constrained by limited operations or large footprints/volume. Here, we showcase a Fourier-based metaprocessor to impart customized highly flexible transfer functions for analog computing upon our single-layer Huygens' metasurface. Basic mathematical operations, including differentiation and cross-correlation, are performed by directly modulating complex wavefronts in spatial Fourier domain, facilitating edge detection and pattern recognition of various image processing. Our work substantiates an ultracompact and powerful kernel processor, which could find important applications for optical analog computing and image processing.

[1] Advanced Microscopy and Instrumentation Research Center, School of Instrumentation Science and Engineering, Harbin Institute of Technology, Harbin 150080, China. [2] Department of Electrical and Computer Engineering, National University of Singapore, Singapore 117583, Singapore. [3] Key Laboratory of Ultra-Precision Intelligent Instrumentation of Ministry of Industry and Information Technology, Harbin Institute of Technology, Harbin 150080, China. [4] Key Laboratory of Micro-Systems and Micro-Structures Manufacturing of Ministry of Education, Harbin Institute of Technology, Harbin 150080, China. [5] State Key Laboratory of Robotics and Systems, Harbin Institute of Technology, Harbin 150001, China. [6] Department of Microwave Engineering, Harbin Institute of Technology, Harbin 150001, China. [7] LEME, UPL, Univ Paris Nanterre, F92410 Ville d'Avray, France. [8] These authors contributed equally: Zhuochao Wang, Guangwei Hu, Xinwei Wang. ✉email: xuminding@hit.edu.cn; zhangkuang@hit.edu.cn; lihaoyu@hit.edu.cn; sburokur@parisnanterre.fr; chengwei.qiu@nus.edu.sg

High-speed and high-efficiency computation is central in many modern technologies. The digital computation based on electron flows in microelectronic circuits relies on analog-to-digital conversions, which generally suffers from high energy consumption, low operation speed, and systematic complexity[1], making it challenging especially for massive data processing. Instead, analog computing directly performs mathematical operations in parallel and thus avoids converting massive data to discretized bits. Traditional analog computers take various forms of mechanical, electronic, or hybrid devices and have large size and slow responses[2–4], hindering their readiness of ultrafast computations. Recently, all-optical analog computing provides an alternative computational platform at the speed of light with low power consumption, and could be integrated into photonic chips with thin and planar profiles as well as enhanced compactness[5], which is thus widely explored as the next-generation computation tools.

Fundamentally, analog optical computation can be implemented in temporal or spatial domain. Time-domain analog calculations process pulse signals from limited input ports[6,7]. In stark contrast, spatial analog computing modulates incident wavefronts in real space, enabling massive and high-throughput parallel operations for required signal-processing tasks such as spatial differentiation[8–10], integration[11] and solving equations[12,13]. Conventional physical architectures of spatial domain analog computers leverage upon the phase accumulation with stacked or series of optical elements[14,15], making the whole system bulky and lossy. Nevertheless, metasurface has been captivated as a popular notion and a promising candidate for highly efficient, compact and ultrathin analog processors[16–18].

For instance, an inverse-designed computational meta-structure is proposed to solve linear integral equations[19].

Therein, the analog operator coupled to waveguides can directly modulate the wave (Fig. 1a), solving Fredholm integral equation in a recursive system. Nevertheless, the feedback mechanism induces the time delay and intrinsic discretization errors in signal sampling, which may also impose demanding nanofabrication requirements in higher frequencies. To directly manipulate analog input waves, Fourier systems with engineered transfer functions are employed to attain the prescribed mathematical kernel. Generically, using the classical optical 4f system, two pieces of graded refractive index lenses are required to perform direct and inverse spatial Fourier transform, while a metasurface in Fourier plane can modulate the spatial frequency (Fig. 1b)[2,20–22]. Such method offers the real-time wave-based processing mechanism but still demands bulky systems in practical implementations, especially when the overall space occupied by the 4f system is counted.

On the other hand, exploiting Green's function (GF) kernel to directly operate on the angular scattering spectrum of impinging waves can bypass those challenges, since it avoids the subblocks of Fourier transform. The successful proposals along this paradigm include, for example, optical differentiation with multi-layered slabs[23], a nonlocal metasurface with Fano response[24], the Kretschmann configuration for polaritons[25], photonic crystal slab[26,27] and air-glass interface for spin Hall effect[28]. However, the scattering spectrum of the meta-atoms for those designs is angle dependent, which necessarily decreases incident-angle range and operation diversity[23] (see Supplementary Note 1 for more details). Hence, we propose a single-layer metasurface-based analog processor (Fig. 1c). Such proposal with restrictions on the fixed input and output focal length overcomes the existing issues of compactness and maneuverability of traditional bulky

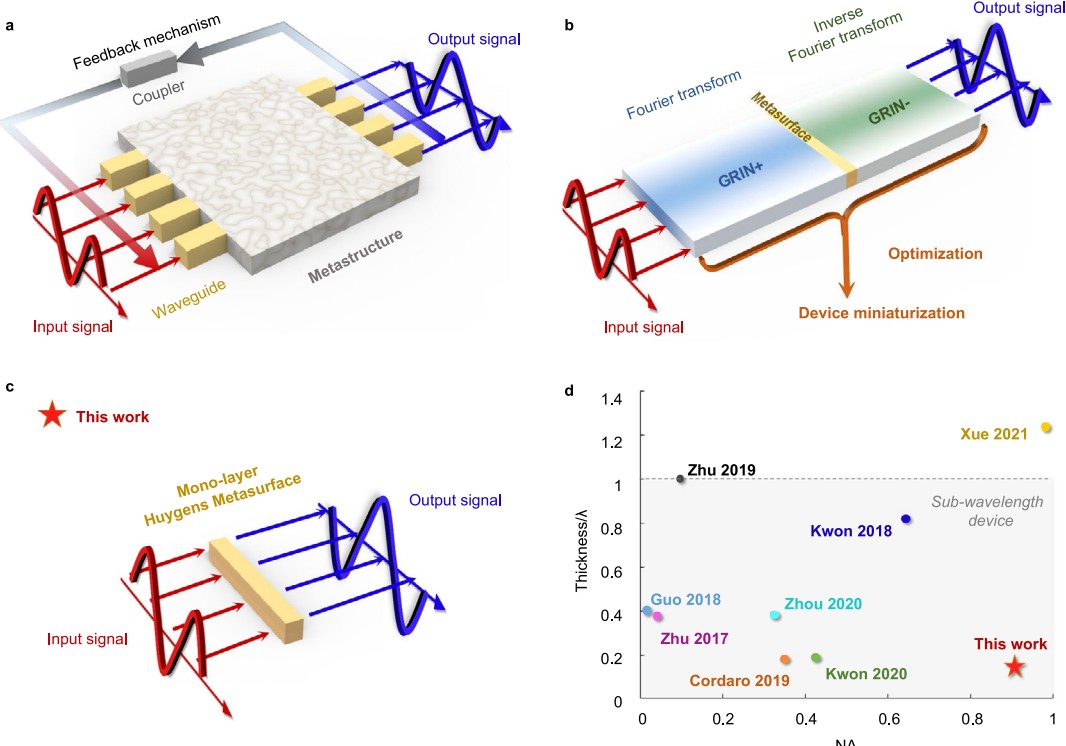

**Fig. 1 Conceptual sketch of representative analog metaprocessors. a** Solving integral equation by a closed-loop network consisting of the metamaterial kernel, directional couplers and in/out waveguides. **b** Spatial analog computing based on the 4f Fourier optics system. (GRIN+ indicates a conventional GRIN, operates as a Fourier transformer. GRIN− has the inverse functionality of the GRIN+, which acts as an inverse Fourier transformer). **c** The proposed single-layer meta-processor in this work for shrinking the bulky Fourier-based designs. **d** Comparison of the proposed differentiator to the recently proposed designs[5,24,26,31-34].

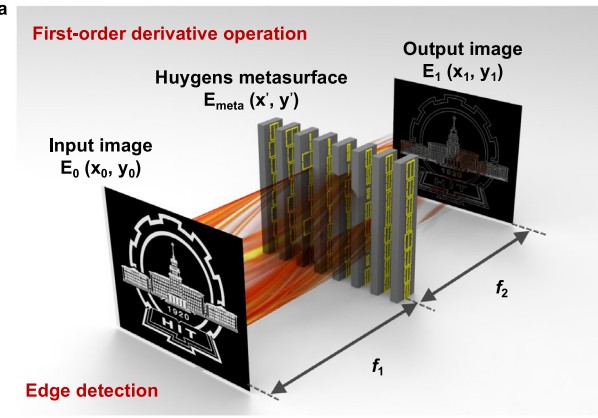

**a** First-order derivative operation

Huygens metasurface
$E_{meta}$ (x', y')

Output image
$E_1$ (x₁, y₁)

Input image
$E_0$ (x₀, y₀)

$f_2$

$f_1$

Edge detection

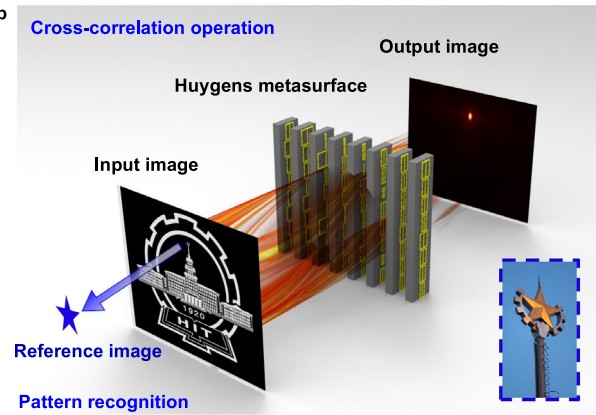

**b** Cross-correlation operation

Output image

Huygens metasurface

Input image

Reference image

Pattern recognition

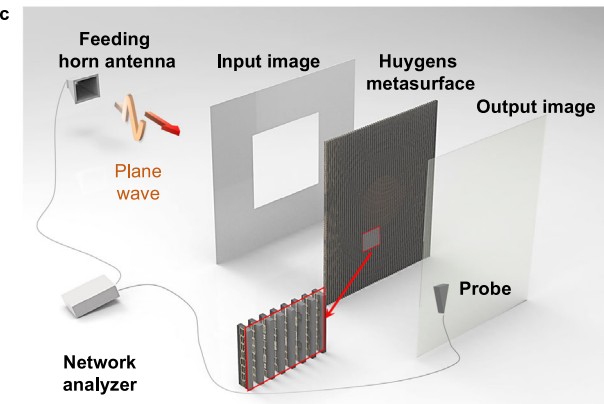

**c**

Feeding horn antenna

Input image

Huygens metasurface

Output image

Plane wave

Probe

Network analyzer

**Fig. 2 Schematic diagram of analog computational system utilizing single-layer Huygens' metasurface. a** Derivative operation and **b** cross-correlation operation performed on the input image. The inset shows the photograph of the star in the badge of HIT, which is defined as the reference image here. **c** The experiment set-up with a zoomed-in view of the fabricated Huygens' metasurface (see Supplementary Note 9 for photographs of the fabricated Huygens' metasurfaces).

Fourier optical devices and shows high flexibility compared to the GF approach (Fig. 1d).

Our analog optical processor is based on single-layer Huygens' metasurface (Fig. 2a, b). By introducing specific phase factor into the complex wavefront profile on the interface, the single-layer meta-structure herein directly tailors the transverse wavevector exerted in spatial Fourier spectrum and hence imposes the customized transfer function for analog image processing, which can essentially compress the typical 4*f* optical system to the 2*f*

structure (Fig. 2c). For this purpose, we use a single-layer Huygens' metasurface, which can facilitate the complete control of wavefronts and amplitudes via judiciously tailored electric and magnetic dipoles[29–31], to realize the desired transfer function required by the versatile implementable kernels of mathematical operations. As the proof of concept, we demonstrated differentiation and cross-correlation for the edge detection and object recognition, in an analog and compact manner. Compared with reported state-of-the-art edge detectors[5,24,27,32–35], our proposed system demonstrates a relatively superior balance between the NA and device dimension (Fig. 1d), thanks to the on-demand electromagnetic control via Huygens' metasurfaces. Our work could enable the real-time and high-throughput parallel computing tasks, overcoming the existing integration issues of traditional bulky Fourier optical devices while performing diverse mathematical operations in contrast to GF kernels. Hence, our proposed miniaturized meta-processor revels high-performance computing and can be readily generalized for tremendous tasks in analog imaging processing[36,37] and computations such as equation solvers[19,38], edge detection of patterns[27,35], optical memory[39], machine learning[40–42] and others.

## Results

**Design of Huygens' metasurface processor.** We firstly discuss the principle of Huygens' metasurfaces as the image processor. As shown in Fig. 2a, the input image, Huygens' metasurface, and the output plane are positioned at $z = -f_1$, $z = 0$ and $z = f_2$ respectively. To avoid forward and inverse Fourier transform subblocks in bulky 4*f* systems, electromagnetic responses of Huygens' metasurface image processor should be elaborately designed. Under the paraxial approximation and in Fresnel regime, the wave in output plane (dubbed as $E_1$), regarding the impinging image ($E_0$) through the Huygens' metasurface ($E_{meta}$), can be expressed:

$$E_1(x_1,y_1) = -\frac{k^2}{4\pi^2 f_1 f_2}\exp[ik(f_1+f_2)]$$
$$\iint\limits_{\Sigma_{meta}} dx'dy'\left\langle \iint\limits_{\Sigma_0} dx_0 dy_0 E_0(x_0,y_0)\exp\left\{\frac{ik}{2f_1}\left[(x'-x_0)^2+(y'-y_0)^2\right]\right\}\right\rangle$$
$$E_{meta}(x_0,y_0)\exp\left\{\frac{ik}{2f_2}\left[(x_1-x')^2+(y_1-y')^2\right]\right\} \tag{1}$$

To directly tailor the spatial Fourier spectrum of the input signal, the aperture function on the Huygens' metasurface is defined as:

$$E_{meta}(x',y') = \exp\left[-\frac{ik}{2f}(x'^2+y'^2)\right]E_H(x',y') \tag{2}$$

where $\frac{1}{f}=\frac{1}{f_1}+\frac{1}{f_2}$ (see Supplementary Note 2 for more details). Here, the phase factor $\exp[-\frac{ik}{2f}(x'^2+y'^2)]$ is introduced to suppress the undesired phase terms for the construction of linear convolution relationship at the output plane. Hence, $E_H$ denotes the transfer function associated with desired mathematical operations in spatial Fourier domain, which plays central roles to modulate spatial spectrum responses. For the specific wavefront impinged on our Huygens' metasurface, the integral in Eq. (1) can be re-evaluated as:

$$E_1(x_1,y_1)=\left\{\frac{k^2}{2\pi f_1 f_2}\exp[ik(f_1+f_2)]\exp\left[\frac{ik}{2f_2}(x_1^2+y_1^2)\right]\right\}$$
$$\times\left\{E_0\left(-\frac{f_1}{f_2}x_1,-\frac{f_1}{f_2}y_1\right)\phi(x_1,y_1)\right\}\odot F\{E_H(x',y')\}[k_x,k_y] \tag{3}$$

where the additional phase factor $\phi(x_1,y_1)=\exp[\frac{ikf_1}{2f_2^2}(x_1^2+y_1^2)]$ and $k_x=\frac{k}{f_2}x_1$ and $k_y=\frac{k}{f_2}y_1$ are spatial frequencies (wavevectors) along $x$ and $y$-axes, respectively. Besides, $\odot$ represents

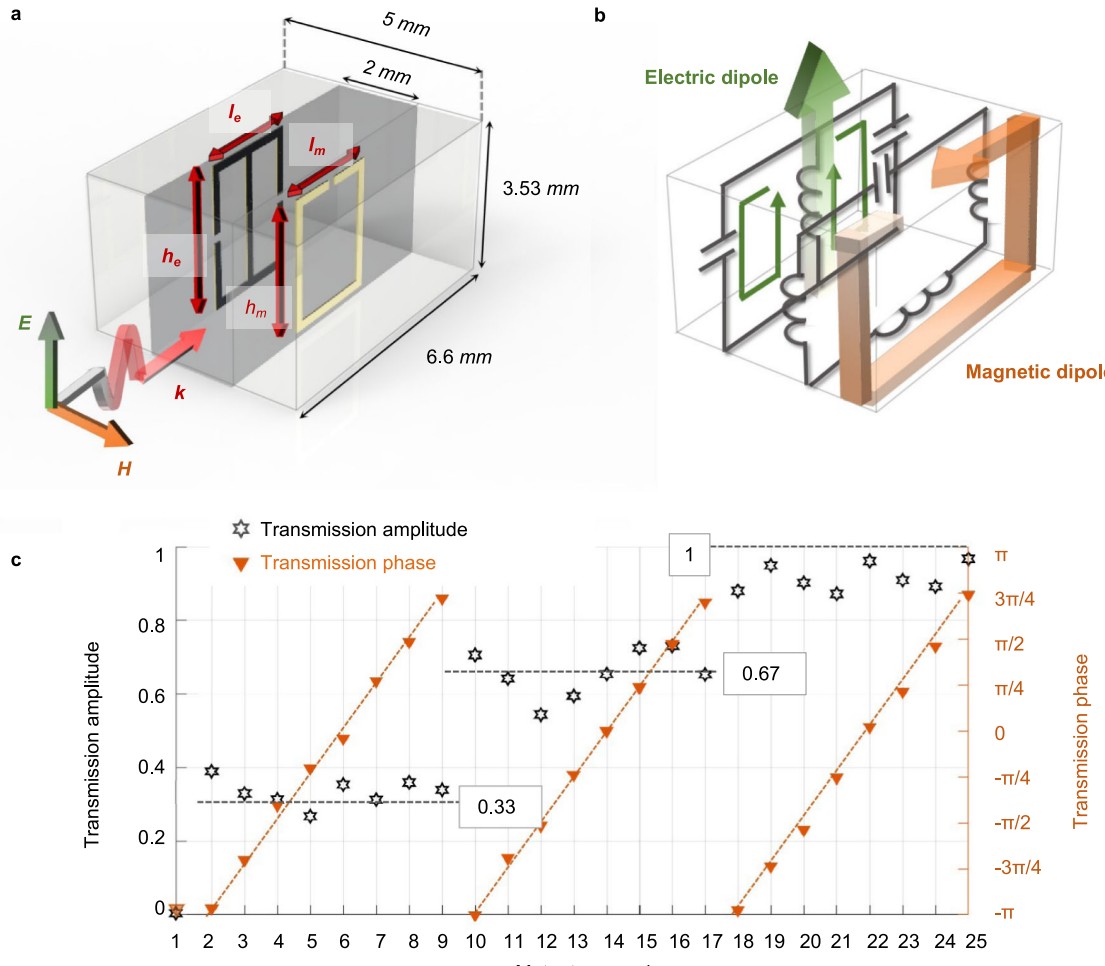

**Fig. 3 Design of Huygens' metasurface processor. a** Geometric structure of the proposed elementary Huygens' meta-atom patterned on both sides of the dielectric substrate ($\varepsilon_r = 3$, $\tan\delta = 0.002$). **b** Surface currents and equivalent electric circuits of the proposed meta-atom. The green line with arrows and orange arrows indicate the surface electric and magnetic current respectively. **c** Simulated transmission amplitude and phase of 25 extracted meta-atoms by modulating the geometrical parameters $l_e$, $h_e$ and $l_m$, $h_m$ at 10 GHz.

two-dimensional convolution operation. Essentially, this equation shows that the output analog signal is the convolution of the inversed and rescaled input image and the spatial frequency-domain $E_H(x', y')$ on Huygens' metasurfaces. Hence, the following ouput images are rotated by $\pi$ about the origin to compare with the input ones. Via applying the convolution theorem $F\{f(x) \odot g(x)\} = F\{f(x)\} \times F\{g(x)\}$ to Eq. (3), the output image in Fourier spectrum can be obtained by multiplying input signals with repeated Fourier transforms of $E_H$, as:

$$F\{E_1(x_1, y_1)\} = \{...\}F\{E_0(x_1, y_1)\phi(x_1, y_1)\} \times F\langle F\{E_H(x', y')\}\rangle$$
$$= \{...\}F\{E_0(x_1, y_1)\phi(x_1, y_1)\} \times E_H(-x', -y')$$

$$(4)$$

Since the definition of transfer function, which is the ratio of the output signal to input signal in Fourier spectrum, is given as $E_H(-x', -y') = \frac{F\{E_1(x_1, y_1)\}}{F\{E_0(x_1, y_1)\phi(x_1, y_1)\}}$, $E_H$ can model the spatial frequency response of the proposed metasurface-based processor quantitively and map the output image for each possible input. For maneuverability without loss of generality, the transfer function of first-order differential operator is $E_H(x) \propto jx$ and for the integrator, transfer function can be described as $E_H(x) \propto \frac{1}{jx}$. Overall, we can superimpose the specific phase factor related with the input and output focal length $\exp[-\frac{ik}{2f}(x'^2 + y'^2)]$ on the

transfer function $E_H$ algorithmically. In this manner, the proposed single-layer Huygens' metasurface can directly implement spatial frequencies for the target output signal, which avoids auxiliary optical elements.

To fulfill the appropriate transfer function, we use the Huygens' metasurface here as the analog processor due to its extreme control of complex wavefronts across just one single-layer rather than multi-layer metasurface designs[29]. By incorporating orthogonal electric and magnetic dipoles with carefully tailored resonant amplitude into meta-atoms on the interface, Huygens' surface introduces abrupt field discontinuities to generate prescribed wavefronts[30] and enables both complete amplitude and phase manipulation with 100% transmission efficiency (see more details in Supplementary Note 3). Here, the metallic Huygens' metasurfaces are designed in microwave regime, where the dissipative losses of metals are negligible. As displayed in Fig. 3a, our Huygens' meta-atoms are composed by two split-ring resonators at both sides[31]. When excited by the incident wave carrying the image information, a magnetic dipole, aligned with incident magnetic field, is induced by the split-ring resonator on one side of substrate (marked in orange in Fig. 3b)[43,44], while the electric dipole is on the other side, in a form of the electric-LC resonator with main surface current flowing along the incident electric field (marked in green in Fig. 3b)[45]. Importantly, the light-matter interaction strength as

well as induced dipole moment can be further altered by optimizing geometrical parameters ($l_e$, $h_e$ and $l_m$, $h_m$). Figure 3c provides the numerical result of 25 optimized Huygens' meta-atoms using the commercial software CST Microwave Studio (see Supplementary Note 3 for more details). The proposed Huygens' meta-atoms can achieve full-coverage modulation of transmission phase quantified into 8 levels and complete transmission amplitude control discretized into 4 levels at 10 GHz, as building blocks of the desired analog image processors. Particularly, when the transmission amplitude of meta-atom equals to 0, its phase shifts contribute no variations on the transmitted wavefront. Therefore, only one Huygens' meta-atom is designed for transmission amplitude equivalent to 0 with transmission phase of $-\pi$. Moreover, the deviations between the simulated transmission coefficient and the predesigned value result from the minimum process tolerance, especially for meta-atoms with the transmission amplitude of 0.67 and 1, which are more sensitive to the offsets of overlaps between the magnetic and electric resonance. Significantly, the control of both amplitude and phase of monochromatic waves enables the diversity of mathematical operations. The single-layer meta-processor enables a large variety of mathematical operations, including spatial shifting, high-order derivative, spectral filtering and other Fourier transform algorithms. In the following, we will show two exemplary important applications of image detection.

**Edge detection operation**. Edge detection is commonly explored to characterize subject boundaries of an image and, in a mathematic language, is reflected in a dramatic change of the derivatives of signals due to the sudden change of objects. The simplest yet powerful way for such implementation is to check the behavior of the first-order derivative operation, which is denoted (with respect to $x$-axis for example):

$$E_H(x') \propto jx' \iff F\{E_H\}[k_x] \propto \frac{d}{dk_x}\delta(k_x) \quad (5)$$

According to Eqs. (3) and (5), the first-order differentiation of the relevant input image $E_0(-x_1)\phi(-x_1)$ in spatial domain maps the multiplication of $jx'$ in spatial Fourier spectrum. By modulating the amplitude and phase distributions of transmitted wavefronts according to Eq. (4), the absolute value of the output complex image given as:

$$|E_1(x_1,y_1)| \propto \left|\frac{d\{E_0(-x_1)ph(-x_1)\}}{dx_1}\right| = \left|\frac{d\left\{E_0(-x_1)\exp\left(\frac{ik}{2f_2}x_1^2\right)\right\}}{dx_1}\right|$$
$$= \sqrt{\left[\frac{dE_0(-x_1)}{dx_1}\right]^2 + \left[\frac{k}{z_1}x_1E_0(-x_1)\right]^2} \approx \frac{dE_0(-x_1)}{dx_1} \quad (6)$$

can be regarded as the processed results under first-order differentiation approximately. The derivation above demonstrates the spatial differentiator along $x$-direction. Similarly, the spatial differentiator along $y$-axis can be implemented by replacing variable $x$ with $y$. Hence, the first-derivative operation along the orthogonal axes can be performed based on the proposed Huygens' metasurfaces with complex wavefronts modulation, as shown in Fig. 4a, c. Figure 2c illustrates the image processing system in microwave regime. A feeding horn antenna launches a quasi-plane wave on the metal sheet with the predefined holes for the generation of the input image at z = $-f_1$. Then, after modulating the wavefronts by Huygens' metasurface at z = 0, the transmitted electric field in the output plane is measured at z = $f_2$ with a near-field probe. As proof-of-concept experiment, the input image is set to be a specific rectangle with length

$l_1 = 225\,mm$ and width $l_2 = 305\,mm$ under the focal-length conditions $f_1 = f_2 = 100\,mm$. Based on Eqs. (2) and (5), the phase and amplitude profiles on Huygens' metasurface are easily designed for edge detection in $x$ and $y$-directions (see more details in Supplementary Note 4). Figure 4b, d displays the theoretical, numerical and measured electric field intensity profiles, which show an excellent agreement. One could clearly see strong signals at the edge location of images, verifying the 1D-edge detection capabilities of the proposed analog image processors based on Huygens' metasurfaces.

Consequently, the 2D-edge and vertex detection, as shown in Fig. 4e, g, can be performed by further processing of the obtained orthogonal 1D-edge information. Firstly, based on the linearity of Fourier transform, the edge information on the $x$- and $y$-axes can be registered simultaneously by superimposing their corresponding transfer functions in spatial Fourier domain. Hence, 2D-edge detection $\left(\left|\frac{dE_0(x_1)}{dx_1} + j\frac{dE_0(y_1)}{dy_1}\right|\right)$ can be achieved by the superposition operation on electric field profiles of Huygens' metasurface along orthogonal axes, as:

$$E_H(x',y') = E_H(x') + jE_H(y') \propto j(x' + jy') \quad (7)$$

Secondly, by extracting the common parts of $x$- and $y$-directions edge data that coexist at the same positions, vertex detection $\left(\left|\frac{dE_0(x_1)}{dx_1} \times \frac{dE_0(y_1)}{dy_1}\right|\right)$ can be implemented through the multiplication on 1D-edge detection Huygens' metasurface profiles along $x$-axis and y-axis, as:

$$E_H(x',y') = E_H(x') \times E_H(y') \propto -x'y' \quad (8)$$

Due to the direct modulation in spatial Fourier spectrum by the Huygens' metasurface, the proposed processor can be optimized by increasing the number of meta-atoms for superior light-gathering ability and resolution (see more details in Supplementary Note 5). Hence, to detect the details of input rectangular image at 100 mm, the number of meta-atoms is set to be $100 \times 140$ to keep NA as high as 0.9. Due to the single-layer structure, the overall thickness of operator is ~$\lambda/6$, which ensures ultra-compactness and integration of our devices. As exhibited in Fig. 4f, h, the simulated and measured electric field intensity profiles are consistent with the theoretical results, demonstrating the 2D-edge and vertex detection functionalities of the proposed structures. Moreover, to validate the maneuverability of our proposed processor across versatile applications, the 2D-edge detection of more complex patterns, including regular pentagon and hexagon, are also implemented by increasing meta-atom number to $120 \times 170$. As illuminated in Fig. 4i, j, the 2D first-order derivative operations are completely performed on both images for edge information extraction. Nevertheless, in all experimental demonstrations, a minor deviation between the theoretical and measured figures can be seen, which is attributed to the limited dimensions of Huygens' metasurfaces and wavefront discretization. The proposed processors could be further improved, and the output edge information can be more uniform via optimizing the phase- and amplitude- quantization (see Supplementary Note 6 for more details). In summary, our single-layer Huygens' metasurface image processor can efficiently implement the edge and vertex detection techniques.

**Cross-correlation operation**. Note that edge detection essentially characterizes the sharp edges of objects. However, such method neglects other important details of the image, such as continuous areas of objects. Therefore, to provide global evaluations of the image, we resort to another image operation—correlation. In signal-processing, cross-correlation is a fundamental operation to

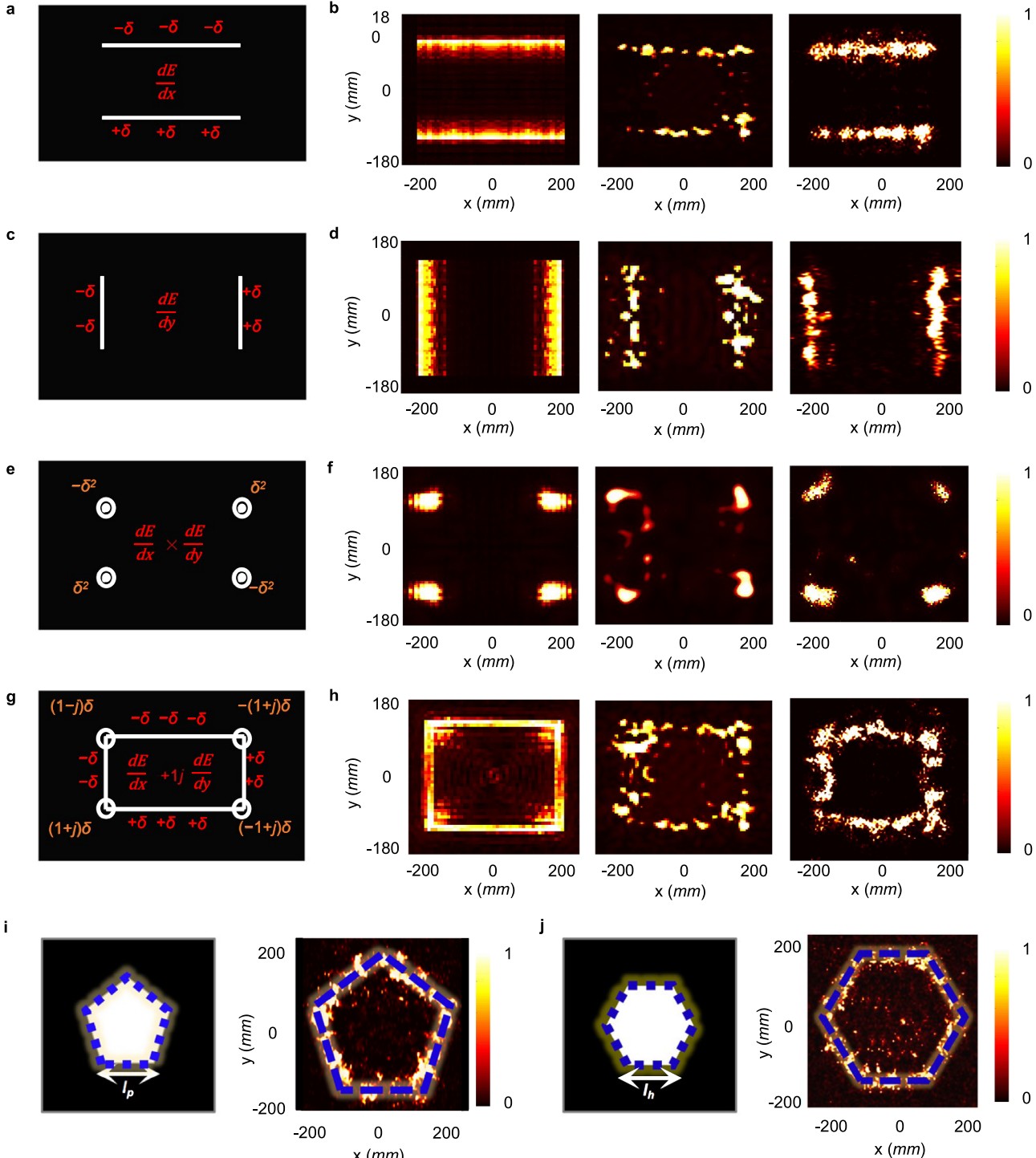

**Fig. 4 The first-order derivative operation. a** Schematic diagram and **b** theoretical calculations (left), numerical simulations (middle) and experimental results (right) of the normalized electric field intensity distribution at the output focal length $f_2 = 100$ mm and 10 GHz working frequency for 1D-edge detection along x-axis. **c** Schematic diagram and **d** theoretical calculations (left), numerical simulations (middle) and experimental results (right) of the normalized electric field intensity distribution at the output focal length $f_2 = 100$ mm and 10 GHz working frequency for 1D-edge detection along y-axis. **e** Schematic diagram and **f** theoretical calculations (left), numerical simulations (middle) and experimental results (right) of normalized electric field intensity distribution at the output focal length $f_2 = 100$ mm and 10 GHz working frequency for vertex detection. **g** Schematic diagram and **h** theoretical calculations (left), numerical simulations (middle) and experimental results (right) of normalized electric field intensity distribution at the output focal length $f_2 = 100$ mm and 10 GHz working frequency for 2D-edge detection. **i** The input image of regular pentagon (left) with $l_p = 210$ mm and output experimental results (right) of normalized electric field intensity distribution at the output focal length $f_2 = 100$ mm and 10 GHz working frequency for 2D-edge detection. **j** The input image of regular hexagon (left) with $l_h = 175$ mm and the output experimental results (right) of normalized electric field intensity distribution at the output focal length $f_2 = 100$ mm and 10 GHz working frequency for 2D-edge detection.

measure the similarity of two images or data arrays:

$$f(x) \otimes g(x) = \int dx' f(x') g^*(x' - x) = f(x) \odot g^*(-x) \quad (9)$$

Here, $\otimes$ represents the cross-correlation operation. Correspondingly, by the complex conjugate processing of $E_H(x)$, as $E_H^*(x) \Longleftrightarrow F^*\{E_H\}[-k_x]$, the convolution relationship between the inversed input image and transfer function in Fourier domain in Eq. (3) can be transformed into cross-correlation operation:

$$\begin{aligned} E_1(x_1, y_1) &\propto E_0(-x_1, -y_1) \odot F^*\{E_H\}[-k_x, -k_y] \\ &= E_0(-x_1, -y_1) \otimes F\{E_H\}[k_x, k_y] \end{aligned} \quad (10)$$

In particular, the autocorrelation is defined as the cross-correlation of a function with itself. Clearly, the maximum of the autocorrelation is located at zero offset (i.e., no position shift), and its value denotes the signal energy, as $f(0) \otimes f(0) = \int dx' |f(x')|^2$. Besides, if $g(x)$ is a shifted of $f(x)$, since:

$$f(x) \otimes g(x) = f(x) \otimes f(x - t) = \int dx' f(x') f^*(x' - t + x) = R_{ff}(x - t) \quad (11)$$

the peak of the cross-correlation operation will move at the same distance. According to this property, the cross-correlation operation can be utilized to look for the same features between the input image and the reference image (patterns to be detected) achieved by the predesigned aperture function on Huygens' metasurface, which is of great potentials in various modern technologies including the auto-driving and face recognition.

To validate our metasurface-based analog computing processor, an input image is selected which contains the square #1 with size $l_{11} = l_{12} = 60$ mm and the square #2 with size $l_{21} = l_{22} = 25$ mm. Particularly, the square #1, part of the input signal, is set to be the reference image, described as the 2D rectangular function $F\{E_H\}[k_x] = rect\left(\frac{k_x}{L_x}\right) rect\left(\frac{k_y}{L_y}\right)$. Hence, to build the reference image physically, the transfer function on Huygens' metasurface is designed in the form as:

$$E_H^*(x', y') = E_H(x', y') \propto \left[ L_x \frac{k}{f_2} \mathrm{sinc}\left(L_x \frac{k}{f_2} x'\right) \right] \times \left[ L_y \frac{k}{f_2} \mathrm{sinc}\left(L_y \frac{k}{f_2} y'\right) \right] \quad (12)$$

where $L_x = l_{11}/2 = 30$ mm, $L_y = l_{12}/2 = 30$ mm and the input and output focal lengths $f_1 = f_2 = 100$ mm. By modulating the aperture function on Huygens' metasurface based on Eq. (12), the peak-value position of the cross-correlation operation identifies the location of the reference image pattern consist in the input image. As shown in Fig. 5, the square #1 can be successfully detected in theoretical, simulated and measured results. Hence, specific patterns can be readily recognized from the analog input data by tailoring the wavefront profiles utilizing Huygens' metasurfaces. To extend applications of the proposed computational metasurface, the input image is further designed to be one-dimensional sequence which contains two squares with width $w_1 = 70$ mm. Then, three reference images which contains one, two or three rectangular patterns with the same width (named as ①,②,③) are implemented in spatial Fourier spectrum respectively utilizing Huygens' metasurfaces (see more details of transfer functions in Supplementary Note 7). As shown in Fig. 5e, the output intensity of electric field indicates the index of geometrical similarity and the location of peaks can identify the regions where input and reference images contain the same pattern features. Hence, our metasurface-based cross-correlators can also perform sequence alignment, offering opportunities for potential signal detection and even DNA sequencing.

The robustness of detection functions enabled by our Huygens' metasurface for cross-correlation processing allows wide applications. Simplify searching a large database, the proposed single-layer correlator can directly map and locate the reference fragment with parallel process to decrease the power consumption and accelerate the speed of signal-processing, which are the major limitations of electrical processing to be tackled. The proposed scheme may benefit the radio-frequency sensing technologies, including locating, recognition and feature estimation. Also, the concept can be readily extended to terahertz and optical frequency ranges (see more details in Supplementary Note 8), which can find potential applications in pattern recognition, information retrieval, single particle analysis, electron tomography, and bioinformatics. Moreover, by adding tunable factors into design of Huygens' meta-atoms, a single correlator can track and monitor versatile objects dynamically.

## Discussion

In summary, we proposed a novel ultracompact analog processing system utilizing Huygens' metasurface to overcome speed and energy limitations of digital devices. Based on Fourier optics, the output electric field intensity demonstrates the convolution of the input signal with the transfer function implemented via a Huygens' metasurface with complex wavefronts modulation on the aperture. Significantly, the proposed single-layer processor can directly modulate the spatial Fourier spectrum without physical lenses for direct and inverse Fourier transform. To modulate the complex wavefront profiles on the interface, 25 Huygens' meta-atoms are exploited for full and independent control of transmission amplitude and phase. By constructing the operator with proposed Huygens' meta-atoms based on Fourier transform properties, differentiation and cross-correlation mathematical operations are performed in proof-of-concept experiments.

The analog processor uses single-layer Huygens' metasurface to directly tailor the spatial Fourier frequencies, overcoming the volume defect and propagation loss of 4f Fourier optics system. Moreover, the proposed scheme presents a recipe of spatial arrangement of 25 static Huygens' meta-atoms at the cost-effective computing load by GF method, allowing practical implementation of optical devices with predesigned and complex angular spectrums. Figure 1d showcases a comparative overview and the advantages of our proposed metasurface-based spatial analog processor have been unanimously depicted. Our results may pave toward high-speed, low-dimension, and high-throughput all-optical computing metasurfaces, empowering advanced technologies in analog-data processing, pattern detection and recognition, and even in bio-medical applications.

## Methods

**Experimental section**. As demonstrated in Fig. 2c, the wavefronts generated from the illuminating horn antenna, propagate through the metal plane $z = -f_1$ and later through the Huygens' metasurface at $z = 0$, and finally ends at $z = f_2$ as the output profiles. The experimental equipment is established in an anechoic-type measurement chamber with the size of $310 \times 310 \times 650$ mm³ bounded by absorbers. For the construction of the input image in microwave regime, a metal plane is perforated for the predesigned pattern apertures to allow incidence with specific shapes propagate through. The feeding horn antenna is placed 650 mm away from the metal plane sample to provide a quasi-planar wave illumination. The metal plane sample is surrounded by absorbing materials to eliminate diffraction on the borders. A fiber optic active antenna, operating as the field-sensing probe, is used to measure the transmitted electric field point by point by step of 2 mm. The feeding horn antenna and the probe are both connected to the vector network analyzer (Agilent 8722 ES) to measure the amplitude and phase of the transmission coefficient (component $S_{21}$ of the scattering matrix).

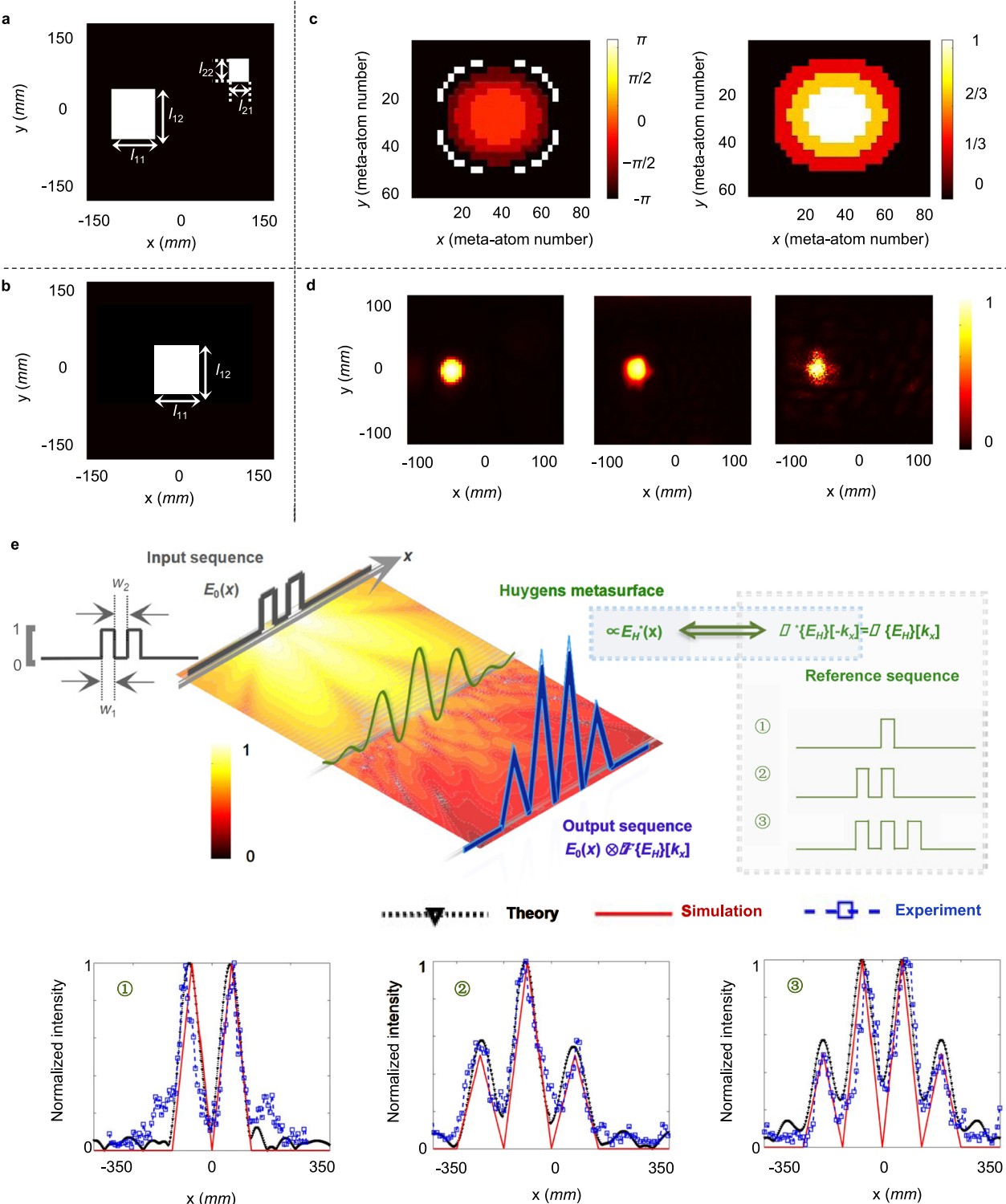

**Fig. 5 Cross-correlation operation. a** The input image consisting of the square #1 with $l_{11} = l_{12} = 60$ mm and the square #2 with $l_{21} = l_{22} = 25$ mm at $f_1 = 100$ mm. **b** The reference image of the square #1. **c** The phase profile (left) and the amplitude distribution (right) on the $60 \times 80$ Huygens' metasurface as the physical block of the reference square #1. **d** Theoretical calculations (left), numerical simulations (middle) and experimental results (right) of the normalized electric field intensity distribution at the output focal length $f_2 = 100$ mm and 10 GHz working frequency for pattern recognition of the square #1. **e** The cross-correlation operation between the input two-square sequence and the reference image containing one, two or three rectangular patterns with $w_1 = 70$ mm (named as ①,②,③ respectively). Besides, The normalized intensity of output electric field along $x$-axis are demonstrated at focal length $f_1 = f_2 = 600$ mm and 10 GHz working frequency for sequence alignment.

## Data availability
All data generated and analyzed are included in the paper and its Supplementary Information.

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

## Acknowledgements
X.D. acknowledges funding from National Key R&D Program of China (2021YFF0603500), Natural Science Foundation of Heilongjiang Province (YQ2021F004). G.H. acknowledges the support from A*STAR under its AME Young Individual Research Grants (YIRG, No. A2084c0172). H.L. acknowledges funding from National Natural Science Foundation of China under Grant No. 61805057. J.L. acknowledges funding from National Natural Science Foundation of China under Grant No. 5197050993. C.W.Q. acknowledges the support by AME Individual Research Grant (IRG) funded by A*STAR, Singapore (Grant No. A2083c0060). C.W.Q. is also supported by a grant (R-261-518-004-720 | A-0005947-16-00) from Advanced Research and Technology Innovation Centre (ARTIC).

## Author contributions
X.D., G.H., and C.W.Q. conceived the idea. Z.W. and X.W. conducted the numerical simulations and fabricated the samples. S.N.B. performed the measurements. Z.W., G.H., X.D., H.L., K.Z., and S.N.B. wrote the manuscript. Q.W., J.L., J.T., and C.W.Q. supervised the overall projects. All the authors analyzed the data and discussed the results. The authors read and approved the final manuscript.

## Competing interests
The authors declare no competing interests.
