## [Peer Review File · Nature Communications]

Single-layer spatial analog meta-processor for imaging processingREVIEWER COMMENTS

Reviewer #1 (Remarks to the Author):

Wang et al showcased single layered Huygens' metasurfaces performing edge detection and cross-correlation operation at microwave regime. With Huygens' meta-atoms which are able to modulate light intensity and phase simultaneously, complicated Fourier optical phenomena have been demonstrated. The key idea of this paper is to realize a transfer function associated with the desired mathematical operation in spatial Fourier domain using a metasurface. The results are interesting, but novelty of the work need to be addressed more to meet the standard of the Nature Communications. The authors can consider following major comments to improve the quality of the works.

1. Huygens' metasurface design

- a. Definition of the aperture function on the metasurface is vague. Where does the phase factor come from? Can it be included the transfer function?
- b. The concept of the transfer function is hard to follow up. Several examples would be helpful for understanding of future readers.
- c. Additional field distribution on Huygens' metasurface should be calculated to display electric and magnetic resonances.
- d. The proposed Huygens' meta-atoms are discretized into 4 phase levels. In the manuscript, a sentence regarding 8 level quantification should be explained further.

2. Edge detection operation

- a. The first-order derivative operation (equation 4) is unclear. It should be described in detail. The transfer function which will be realized by metasurfaces should be derived. What are the intensity and phase map?
- b. Additional simulation and experimental results are needed which can show that the complex amplitude and phase modulation can provide better optical performance rather than phase only modulation.
- c. Experimental results of the first-order derivative operation should be developed further. Could you try such function with more sophisticated image rather than simple line and vertex imaging?

3. Cross-correlation operation

a. The cross-correlation operation (equation 9) is unclear. It should be described in detail. The transfer function which will be realized by metasurfaces should be derived. What are the intensity and phase map?

b. Experimental results of the cross-correlation operation should be developed further. Could you try such function with more sophisticated image rather than simple line and vertex imaging?

4. Technology review

a. The authors provided technology review in figure 1d in terms of thickness and numerical aperture of the device. However, in this paper there is no clear statement or analysis on numerical aperture. Please provide appropriate additional simulation and experimental results in this regard.

Reviewer #2 (Remarks to the Author):

An analog computer is a device that uses continuous variations in a given physical phenomenon to perform some certain calculations or processing tasks. The digital computer based on electron flows in microelectronic circuits relies on analog-to-digital conversions, which generally suffers from high energy consumption and low operation speed, making it challenging for massive data processing. In recent years, there are considerable interest in analogue computing in the context of metamaterials research, since the subwavelength structures could implement computing functionalities by leveraging light propagation in suitably engineered artificial photonic materials. This solution enables ultrafast speeds, low loss, subwavelength, and parallel operations, holding the promise to overcome the aforementioned challenges. The manuscript by Wang et al proposed a Fourier-based transfer functions for analog computing upon single-layer Huygens' metasurface. Some basic mathematical operations are performed by modulating complex wavefronts in spatial Fourier domain. There are several issues in the paper.

1. The title of the paper should be specified, since it looks more like a title of review paper. In particular, it seems to me that the similar topic has been extensively reported by some published papers (see references 20-22 and 32-34). Moreover, some results in the present manuscript have also been partially reported elsewhere. For example, the edge detection with vortex phase has been reported in previous published paper (see reference 21), although the Fourier-based transfer functions have not been mentioned in that paper.

2. In the first paragraph of page 6, the authors write "Fig. 4b, d display the theoretical, numerical and measured electric-field intensity profiles, which show an excellent agreement. One could clearly see strong signals at the edge location of images, verifying the 1D-edge detection capabilities of the proposed analog image processors based on Huygens' metasurfaces." In surprise, it is almost impossible to see the edge enhanced image from the figures (especially in simulations and experimental results). Therefore, I think their claim is, to some extent, exaggerated and misleading.

3. In the proposed scheme, the authors attempt to realize the transfer function by manipulating the wavefront with a single-layer Huygens metasurface. It is known that the edge detection can be evaluated by the spectrum transfer function. However, the absence of spectrum transfer function casts a doubt on the validity of claimed results of the paper.

All of these make the whole protocol doubtful.

Reviewer #3 (Remarks to the Author):

The authors reported a revised analog processor based on the spatial Fourier filtering method using metasurfaces. Compared to traditional 4f system, a single-layer metasurface is used to achieve the function in a more compact 2f system. Comparing to Green's function method, it has more design freedom and flexibility. Optical analog computing using meta-optics is an interesting and promising research field. The work made a good contribution to the development of this field and showed a feasible way of reducing the size of the meta-optics system. Below are some points should be addressed by the authors:

1) The novelty and merits of the work need to be justified clearly. One key novelty, as the authors stated, "Hence, we propose a single-layer metasurface-based analog processor that essentially compresses the typical 4f optical system into the 2f structure." However, those reference works listed by the authors, like Zhou 2020 and Cordano 20219, are really "single-layer" devices which in principle impose no restraints upon the object and image positions, while this work is actually a "2f" device of which the input plane and output plane should be put at the positions of -f and f. In addition, the authors picked the thickness and NA as the two figure of merits to benchmark with other works in Figure 1D. For metasurfaces like that of Zhou 2020 and Cordano 20219, both thickness and NA can be easily designed in the reasonable ranges. Figure 1D doesn't help to justify the novelty and impact of the work.

2) The designs are demonstrated in the microwave frequency, which are usually expected to have more satisfactory results in both simulation and experiments than in optical frequency considering the better fabrication accuracy, less materials limitation and even more design freedom. However, the results showed in Figure 4 are far from satisfactory, even not comparing with other reports but comparing with its own theoretical results. For example, the simulation results in Figure 4 b and d are all broken spots instead of lines showed in Figure 4 a and c. Actually, the simulated result in Figure 4d resembles more of the theoretical result for vortex detection in Figure 4e instead of that for 1D edge detection along y-axis in Figure 4c. These project poor quality to the work.

1) Even though NA is used as one of the two merits for benchmarking, it is not discussed in the main text except appearing in Figure 1d. The authors shall give some elaborations of the importance and difficulties of achieving such a NA and the calculation of the NA in this work.

- 2) Please indicate the resolution of this device, which is an important indicator of the analog processor.
- 3) Figure 1a and b look very nice. Why the authors didn't use them such as the institution's logo to demonstrate the edge detection and pattern recognition? That would make the illustration and real work more consistent and more attractive. Are there any practical challenges to apply the methodology described in the work to a more complex object than e.g. the simple square showed in Figure 5?
- 4) In Figure 3c, the non-zero amplitude of the meta-atoms are fluctuating or deviating a lot from the set value in the dotted line, especially for amplitude of 0.67 and 1. What is the reason?
- 5) Photos or images of the fabricated meta-optics are highly recommended to be shown in the manuscript.
- 6) For Eqn. 10, please check if the middle part of the equation is correct and introduce $R_{ff}(x-t)$ in the text?
- 7) In Figure 5c, seems only central part of the meta-atoms have non-zero amplitude. If that is true, there is no need to design the phase profile for the meta-atoms outside the central part?

Response to the Reviewer’s Comments on the manuscript NCOMMS-21-23665 entitled “*Single-layer spatial analog meta-processor for imaging processing*” submitted to Nature Communications

Firstly, we would like to thank the editor and reviewers for their careful reviewing of our work and for their constructive comments. We are greatly encouraged by the agreement shared by all reviewers about the interest and impact of our work. We are also thankful for their inspiring comments, which have allowed us to further improve the manuscript, both in terms of the theoretical and experimental investigation and of the scientific rigor. Secondly, we would like to apologize for the late response, since the extra experiments took us a much longer time due to the pandemic.

In the following, we provide a point-to-point response to all reviewers’ questions and comments. With this revision, we also attach the revised manuscript and supplementary information addressing the reviewers’ comments. We do hope that the editors and reviewers may appreciate our extended efforts to improve the paper following their suggestions and may find our revised manuscript an exciting contribution to the broad scientific community in optics and optical computations.

Reviewer #1: Wang et al showcased single layered Huygens’ metasurfaces performing edge detection and cross-correlation operation at microwave regime. With Huygens’ meta-atoms which are able to modulate light intensity and phase simultaneously, complicated Fourier optical phenomena have been demonstrated. The key idea of this paper is to realize a transfer function associated with the desired mathematical operation in spatial Fourier domain using a metasurface. The results are interesting, but novelty of the work need to be addressed more to meet the standard of the Nature Communications. The authors can consider following major comments to improve the quality of the works.

Response: The authors thank the reviewer for the positive remarks, which encouraged us a lot. Following the reviewer’s helpful suggestions, we have revised the paper both in theoretical and experimental perspectives, which have greatly improved the current manuscript. Here we summarize the novelty and technical improvement of this work, while the detailed point-to-point responses can be found following this general statement. With this, we hope our significantly revised manuscript could satisfy the stringent standard in publication of Nature Communications.

The novelty of this paper, as now significantly reinforced according to Reviewer’s suggestions, is analyzed in Text S1 of SI , as

“ Text S1 The benchmark table of metastructure-based analog processors

To address the novelty and significance, the benchmarks of our work in contrast to other categories of up-to-date metastructure-based analog processors are listed in terms of working mechanism, device design, device dimension, computing throughput, function diversity, as listed in Table S1. Particularly, according to the digital throughput measured in bits per second (bit/s) , the throughput of analog processor here is evaluated by the processing range of input data per unit of time.

Firstly, both the GF kernel and our proposed processor reveal a distinct advantage in device integration for the single-layer spectrum modulation within sub-wavelength scale. Secondly, compared with the increase of waveguide number with more complicated inverse design of metastructure and the device optimization of GF kernel for wider modulation range of incident angle, the computing throughput can boost enormously by simply increasing the metaatom number of $4f$ Fourier system and our proposed metasurface. Finally, compared with other three methods, the angle-dependent response of GF kernel is suitable for limited Fourier-domain operations (such as derivative and integral). For instance, the differentiation is transformed into the easier multiplication in angular-scattering spectrum, as $E_H(k_x) \propto jk_x$, which can be implemented by the mechanism of Fano resonance or surface plasmon polariton (SPP). However, for the mathematical function with drastic fluctuations, such as the cross-correlation in our work as $E_H(k_x, k_y) \propto \left[L_x \frac{k}{f_2} \text{sinc}(L_x k_x) \right] \times \left[L_y \frac{k}{f_2} \text{sinc}(L_y k_y) \right]$, the difficulties in angle-dependent metaatom design will be greatly increased. Overall, our proposed meta-processor renders a high level of aggregation with respect to the integration, throughput, function diversity. ”

Table S1. Benchmarks of the recently proposed metastructure-based analog processors. The cases highlighted in pale green color are the best implementable computing performances with specific methods compared with others.

	The inverse-designed computational metastructure	$4f$ Fourier system	Green's Function (GF) kernel	Our work
Working mechanism	Spatial-domain calculation	Spatial-frequency-domain calculation	Angular-scattering spectrum calculation	Spatial-frequency-domain calculation
Device design	Inverse-designed metamaterial structure	Two graded refractive index (GRIN) lenses and metasurface	Angle-dependent device	Single-layer metasurface
Device dimension/thickness	Several wavelength scale	Three layers	Single layer	Single layer
Function diversity	Versatile spatial-domain function types	Versatile Fourier-domain function types	Basic Fourier-domain operations	Versatile Fourier-domain function types

As the conclusion of the novelty of our work, the relevant discussions are now added and modified in the Introduction section, as: “By introducing specific phase factor into the complex wavefront profile on the interface, the single-layer meta-structure herein directly tailors the transverse wavevector exerted in spatial Fourier spectrum and hence imposes the customized transfer function for analog image processing, which can essentially compress the typical $4f$ optical system to the $2f$ structure”. Moreover, the advantages has been highlighted and added in the text, as”Our work could enable the real-time and high-throughput parallel

computing tasks, overcoming the existing integration issues of traditional bulky Fourier optical devices while performing diverse mathematical operations in contrast to GF kernels. Hence, our proposed miniaturized meta-processor reveals high-performance computing and can be readily generalized for tremendous tasks in analog imaging processing [35,36] and computations such as equation solvers [19,37], edge detection of patterns [26,34], optical memory [38], machine learning [39-41] and others. ”

1. Huygens’ metasurface design

a. Definition of the aperture function on the metasurface is vague. Where does the phase factor come from? Can it be included the transfer function?

Response: As mentioned in this comment, the definition of the aperture function on metasurface is not discussed elaborately in the main text. Significantly, the introduction of phase factor will contribute to the direct modulation of spatial Fourier spectrum of the input image by transfer function E_H on Huygens’ metasurface, essentially reducing previous $4f$ system to $2f$ system and offering the remarkable compactness of optical computational platforms. For clarify and conciseness, the derivation procedure of the aperture function on Huygens’ metasurface, is analyzed and added in the Text S2 section of Supplementary Information.

Changes being made to the manuscript:

In the SI, we have added the following discussions.

“Text S2 Derivation of the aperture function on Huygens’ metasurface processor

Under paraxial approximation and in Fresnel regime, the wave in output plane (dubbed as E_1), regarding the impinging image (E_0) through the Huygens’ metasurface (E_{meta}), can be expressed as

$$\begin{aligned}
 E_1(x_1, y_1) &= -\frac{k^2}{4\pi^2 f_1 f_2} \exp[ik(f_1 + f_2)] \iint_{\Sigma_{meta}} dx' dy' \left\langle \iint_{\Sigma_0} dx_0 dy_0 E_0(x_0, y_0) \exp\left\{\frac{ik}{2f_1}[(x' - x_0)^2 + (y' - y_0)^2]\right\} \right\rangle \\
 &\quad E_{meta}(x', y') \exp\left\{\frac{ik}{2f_2}[(x_1 - x')^2 + (y_1 - y')^2]\right\} \\
 &= -\frac{k^2}{4\pi^2 f_1 f_2} \exp[ik(f_1 + f_2)] \exp\left(\frac{ik}{2f_2} x_1^2 + \frac{ik}{2f_2} y_1^2\right) \iint_{\Sigma_0} dx_0 dy_0 E_0(x_0, y_0) \exp\left(\frac{ik}{2f_2} x_0^2 + \frac{ik}{2f_2} y_0^2\right) \\
 &\quad \iint_{\Sigma_{meta}} dx' dy' E_{meta}(x', y') \exp\left[ik\left(\frac{x_0}{f_1} + \frac{x_1}{f_2}\right)x' + ik\left(\frac{y_0}{f_1} + \frac{y_1}{f_2}\right)y'\right] \exp\left\{\frac{ik}{2}\left(\frac{1}{f_1} + \frac{1}{f_2}\right)[x'^2 + y'^2]\right\} \quad (S1)
 \end{aligned}$$

From Equation S1, to construct the Fourier transform function in form of $\exp[ikx' +iky']$, the quadratic term $\exp\left\{\frac{ik}{2}\left(\frac{1}{f_1} + \frac{1}{f_2}\right)[x'^2 + y'^2]\right\}$ should be eliminated by introducing the concave-lens phase factor $\exp\left[-\frac{ik}{2f}(x'^2 + y'^2)\right]$ on the Huygens’ metasurface aperture, where

$\frac{1}{f} = \frac{1}{f_1} + \frac{1}{f_2}$. Hence, the the aperture function on the Huygens’ metasurface is defined as

$$E_{meta}(x', y') = \exp\left[-\frac{ik}{2f}(x'^2 + y'^2)\right] E_H(x', y') \quad (S2)$$

where $\frac{1}{f} = \frac{1}{f_1} + \frac{1}{f_2}$. Then, equation S1 can be derived as:

$$\begin{aligned} E_1(x_1, y_1) &= -\frac{k^2}{4\pi^2 f_1 f_2} \exp[ik(f_1 + f_2)] \exp\left(\frac{ik}{2f_2} x_1^2 + \frac{ik}{2f_2} y_1^2\right) \iint_{\Sigma_0} dx_0 dy_0 E_0(x_0, y_0) \exp\left(\frac{ik}{2f_2} x_0^2 + \frac{ik}{2f_2} y_0^2\right) \\ &\quad \iint_{\Sigma_{meta}} dx' dy' E_H(x', y') \exp\left[ik\left(\frac{x_0}{f_1} + \frac{x_1}{f_2}\right)x' + ik\left(\frac{y_0}{f_1} + \frac{y_1}{f_2}\right)y'\right] \\ &= -\frac{k^2}{2\pi f_1 f_2} \left\{ \exp[ik(f_1 + f_2)] \exp\left(\frac{ik}{2f_2} x_1^2 + \frac{ik}{2f_2} y_1^2\right) \right\} \iint_{\Sigma_0} dx_0 dy_0 E_0(x_0, y_0) \exp\left(\frac{ik}{2f_2} x_0^2 + \frac{ik}{2f_2} y_0^2\right) \\ &\quad \mathcal{F}\{E_H(x', y')\} \left[\frac{k}{f_1}(x_0 - \tilde{x}_1), \frac{k}{f_1}(y_0 - \tilde{y}_1) \right] \\ &= -\frac{k^2}{2\pi f_1 f_2} \{ \dots \} \left\{ E_0(\tilde{x}_1, \tilde{y}_1) \exp\left(\frac{ik}{2f_2} \tilde{x}_1^2 + \frac{ik}{2f_2} \tilde{y}_1^2\right) \right\} \odot \mathcal{F}\{E_H(x', y')\} \left[-\frac{k}{f_1} \tilde{x}_1, -\frac{k}{f_1} \tilde{y}_1 \right] \quad (S3) \end{aligned}$$

where $\tilde{x}_1 = -\frac{f_1}{f_2} x_1$, $\tilde{y}_1 = -\frac{f_1}{f_2} y_1$ and \odot represents two-dimensional convolution operation. Via applying the convolution theorem,, described as $\mathcal{F}\{f(x) \odot g(x)\} = \mathcal{F}\{f(x)\} [k_x] \times \mathcal{F}\{g(x)\} [k_x]$, the output image in Fourier spectrum can be obtained by multiplying input signals with repeated Fourier transforms of E_H , as $\mathcal{F}\langle \mathcal{F}\{E_H(x', y')\} \rangle = E_H(-x', -y')$. Hence, as the ratio of the output signal and input signal in Fourier spectrum, E_H acts as transfer function and x' and y' indicate the spatial Fourier frequencies (wavevectors). Huygens' metasurface can directly modulate the spatial Fourier spectrum through E_H for predesigned analog processing. For direct expression of formula in the main text, Equation (S3) can be described as:

$$\begin{aligned} E_1(x_1, y_1) &= \left\{ \frac{k^2}{2\pi f_1 f_2} \exp[ik(f_1 + f_2)] \exp\left[\frac{ik}{2f_2}(x_1^2 + y_1^2)\right] \right\} \left\{ E_0\left(-\frac{f_1}{f_2} x_1, -\frac{f_1}{f_2} y_1\right) \phi(x_1, y_1) \right\} \\ &\quad \odot \mathcal{F}\{E_H(x', y')\} [k_x, k_y] \quad (S4) \end{aligned}$$

where the additional phase factor $\phi(x_1, y_1) = \exp\left[\frac{ikf_1}{2f_2^2}(x_1^2 + y_1^2)\right]$ and

$k_x = \frac{k}{f_2} x_1$ and $k_y = \frac{k}{f_2} y_1$. Overall, by superimposing the specific phase factor related with the

input and output focal length $\exp\left[-\frac{ik}{2f}(x'^2 + y'^2)\right]$ on the transfer function E_H

algorithmically, the proposed Huygens' metasurface can directly manipulate spatial Fourier frequencies for the target outputs with single-layer structure.”

b. The concept of the transfer function is hard to follow up. Several examples would be helpful for understanding of future readers.

Response: The authors thank the reviewer for the important comment. The definition, functionality and examples of transfer function are clarified and added in the main text. Equation (3) in the main text indicates the convolution relationship between the input image

with the phase factor $\phi(x_1, y_1) = \exp\left[\frac{ikf_1}{2f_2}(x_1^2 + y_1^2)\right]$ and $E_H(x', y')$ on Huygens' metasurfaces. Then, the transfer function is introduced to describe the modulation in spatial Fourier spectrum utilizing the proposed Huygens' metasurface.

Changes being made to the manuscript:

The relevant discussions have been added in the main text, as “Via applying the convolution theorem $\mathcal{F}\{f(x) \odot g(x)\} = \mathcal{F}\{f(x)\} \times \mathcal{F}\{g(x)\}$ to Equation (3), the output image in Fourier spectrum can be obtained by multiplying input signals with repeated Fourier transforms of E_H , as

$$\begin{aligned} \mathcal{F}\{E_1(x_1, y_1)\} &= \{\dots\} \mathcal{F}\{E_0(x_1, y_1)\phi(x_1, y_1)\} \times \mathcal{F}\{\mathcal{F}\{E_H(x', y')\}\} \\ &= \{\dots\} \mathcal{F}\{E_0(x_1, y_1)\phi(x_1, y_1)\} \times E_H(-x', -y') \end{aligned} \quad (4)$$

Since the definition of the transfer function, which is the ratio of the output signal to the input

one in Fourier spectrum, is given as $E_H(-x', -y') = \frac{\mathcal{F}\{E_1(x_1, y_1)\}}{\mathcal{F}\{E_0(x_1, y_1)\phi(x_1, y_1)\}}$, E_H can model

the spatial-frequency response of the proposed metasurface-based processor quantitatively and map the output image for each possible input. For maneuverability without loss of generality, the transfer function of first-order differential operator is $E_H(x) \propto jx$ and for the integrator,

transfer function can be described as $E_H(x) \propto \frac{1}{jx}$.”

c. Additional field distribution on Huygens' metasurface should be calculated to display electric and magnetic resonances.

Response: The authors thank the reviewer for the important comment. The surface current simulated by the commercial software CST Microwave Studio is added in Text 2 of SI to validate the electric and magnetic resonances, as

“The proposed Huygens' meta-atoms working at 10 GHz are simulated using the commercial software CST Microwave Studio and the results are depicted in Fig. S1. The split-ring resonator operates as the magnetic dipole with an induced surface current flowing in a loop. On the other side of the substrate, the electric-LC resonator plays the role of the electric dipole with the main surface current flowing along the incident polarized electric field direction, since the capacitive currents are approximately equivalent and flow in opposite direction with respect to the inductor. The magnetic field introduced by the current loops are offset. Therefore, by appropriately changing the length of the resonators, surface electric and magnetic impedance can be adjusted to achieve the desired transmission coefficient.”

Figure S1. The simulated surface current distribution and the corresponding effective electric circuits on the metallic patches.

d. The proposed Huygens' meta-atoms are discretized into 4 phase levels. In the manuscript, a sentence regarding 8 level quantification should be explained further.

Response: The authors thank the reviewer for the comment. Actually, the complex wavefront is quantized to eight-level phase and four-level amplitude modulation. Concretely, by modulating the geometrical parameters of electric and magnetic dipoles, the proposed Huygens' metasurface can manipulate the complex transmission coefficient. Significantly, the amplitude is quantified into 8 levels and the phase is discretized into 4 levels. Besides, the correlative discussion is described in the paper, as “The proposed Huygens' meta-atoms can achieve full-coverage modulation of transmission phase quantified into 8 levels and transmission amplitude control discretized into 4 levels at 10 GHz, as building blocks of the desired analog image processors.”

2. Edge detection operation

a. The first-order derivative operation (equation 4) is unclear. It should be described in detail. The transfer function which will be realized by metasurfaces should be derived. What are the intensity and phase map?

Response: The authors thank the reviewer for the comment. According to the response 1b, the

transfer function $E_H(-x', -y') = \frac{\mathcal{F}\{E_1(x_1, y_1)\}}{\mathcal{F}\{E_0(x_1, y_1)\phi(x_1, y_1)\}}$ act as the modulation function in

spatial Fourier domain implemented by Huygens' metasurface, where the phase factor

$\phi(x_1, y_1) = \exp\left[\frac{ikf_1}{2f_2^2}(x_1^2 + y_1^2)\right]$. Hence, according to the properties of Fourier transform, the

first-order differentiation on the input image $E_0(x_1, y_1)\phi(x_1, y_1)$ in spatial domain maps the multiplication in spatial Fourier spectrum, as

$$E_H(x') \propto jx' \Leftrightarrow \mathcal{F}\{E_H\}[k_x] \propto \frac{d}{dk_x} \delta(k_x)$$

Moreover, the relevant discussion have been added in the main text, as “ According to Equation (3) and (5), the first-order differentiation of the relevant input image $E_0(-x_1)\phi(-x_1)$ in spatial domain maps the multiplication of jx' in spatial Fourier spectrum. ”

Besides, since the first-order derivative operation is performed for x-axis edge detection, y-axis edge detection, vertex detection and 2D edge detection, 4 pairs of amplitude and phase maps should be demonstrated. For conciseness of main text, the analyses of the intensity and phase distribution on Huygens' metasurface (including the relevant formulas and maps) have been described in Text S4 of Supplementary Information.

b. Additional simulation and experimental results are needed which can show that the complex amplitude and phase modulation can provide better optical performance rather than phase only modulation.

Response: The authors thank the reviewer for the comment. The control of amplitude and phase of monochromatic waves enables the diversity of mathematical operations. Both the first-order derivative and cross-correlation operation need the complex wavefront control. On the other hand, the wavefront discretization can influence the optical performance. And the relevant discussions have been added in Text S5 section of Supplementary Information, as

“Text S5 Influence of phase and amplitude discretization on the performance of 2D-edge detection

The influence of phase and amplitude discretization on the Huygens' metasurface-based processor is analyzed from the examples of 2D-edge detection. As shown in Fig. S3, the processor with nearly smooth phase and amplitude modulations gives the sharpest feature of the edge. With the decrease of quantization level of complex-wavefront modulation, the edges are still apparent, but show wider peaks, more significant side lobes and more background noises, especially under two-level phase modulation and four-level amplitude modulation (2PM-4AM) and 8PM-2AM. The results validate the fact that the output performance can be improved by increasing the phase and amplitude quantization level. ”

Figure S3. Influence of phase and amplitude discretization on the performance of 2D-edge detection. (a) The output electric field intensity distribution utilizing Huygens' metasurface with different phase modulation and amplitude modulation levels: 2PM-4AM, 4PM-4AM, 8PM-4AM, 16PM-4AM respectively for the analysis of phase quantization. (b) The output electric field intensity distribution utilizing Huygens' metasurface with 8PM-2AM, 8PM-4AM, 8PM-8AM, 8PM-16AM respectively for the analysis of amplitude quantization. The cases highlighted in pale blue color in the (a) and (b) are the same 8PM-4AM quantization configuration selected in the main text.

c. Experimental results of the first-order derivative operation should be developed further. Could you try such function with more sophisticated image rather than simple line and vertex imaging?

Response: The authors thank the reviewer for the comment. To develop the proposed processor, we have done attempts to improve the experimental results of the first-order derivative operation on input rectangle and added new validation of the same function on more sophisticated images.

Firstly, experimental results of x-axis edge detection, y-axis edge detection, vertex detection and 2D edge detection on rectangular image have been optimized by increasing the number of meta-atoms from 50×90 to 100×140 and NA from 0.78 to 0.9. As shown in Fig. R1.1, the edge width and background noise are significantly decreased when a larger metasurface is used.

Figure R1.1. The comparison diagram of normalized electric field distribution for x-axis edge detection, y-axis edge detection, vertex detection and 2D edge detection utilizing 50×90 (left) and 100×140 (right) Huygens' metasurface.

Furthermore, more sophisticated image patterns (such as regular pentagon and hexagon) are processed by 120×170 Huygens' metasurface for 2D-edge detection as the proof-of-concept experiments. The relevant discussions have been added in the main text, as

“Moreover, to validate the maneuverability of our proposed processor across versatile applications, the 2D-edge detection of more complex patterns, including regular pentagon and hexagon, are also implemented by increasing meta-atom number to 120×170 . As illuminated in **Fig. 4i, j**, the 2D first-order derivative operations are completely performed on both images for edge information extraction.”

Figure 4. The first-order derivative operation. (a) Schematic diagram and (b) Theoretical calculations (left), numerical simulations (middle) and experimental results (right) of the normalized electric field intensity distribution at the output focal length $f_2 = 100$ mm and 10 GHz working frequency for 1D-edge detection along x -axis. (c) Schematic diagram and (d) Theoretical calculations (left), numerical simulations (middle) and experimental results (right) of the normalized electric field intensity distribution at the output focal length $f_2 = 100$ mm and 10 GHz working frequency for 1D-edge detection along y -axis. (e) Schematic diagram and (f) Theoretical calculations (left), numerical simulations (middle) and experimental results (right) of normalized electric field intensity distribution at the output focal length $f_2 = 100$ mm and 10 GHz working frequency for vertex detection. (g) Schematic diagram and (h) Theoretical calculations (left), numerical simulations (middle) and experimental results (right) of normalized electric field intensity distribution at the output focal length $f_2 = 100$ mm and 10 GHz working frequency for 2D-edge detection. (i) The input image of regular pentagon (left) with $l_p = 210$ mm and output experimental results (right) of normalized electric field intensity distribution at the output focal length $f_2 = 100$ mm and 10 GHz working frequency for 2D-edge detection. (j) The input image of regular hexagon (left) with $l_h = 175$ mm and the output experimental results (right) of normalized electric field intensity distribution at the output focal length $f_2 = 100$ mm and 10 GHz working frequency for 2D-edge detection.

3. Cross-correlation operation

a. The cross-correlation operation (equation 9) is unclear. It should be described in detail. The transfer function which will be realized by metasurfaces should be derived. What are the intensity and phase map?

Response: The authors thank the reviewer for the comment. The relationship between the cross-correlation operation and transfer function is derived and clarified in the main text.

For the cross-correlation operation, the relationship between the convolution and cross-correlation is first derived in the main text, as

$$f(x) \otimes g(x) = \int dx' f(x') g^*(x' - x) = f(x) \odot g^*(-x)$$

Hence, for transforming the equation (3) into the cross-correlation operation, $E_H(x)$ should be complex conjugate, as $E_H^*(x) \Leftrightarrow \mathcal{F}^*\{E_H\}[-k_x]$, since

$$E_1(x_1, y_1) \propto E_0(-x_1, -y_1) \odot \mathcal{F}^*\{E_H\}[-k_x, -k_y] = E_0(-x_1, -y_1) \otimes \mathcal{F}\{E_H\}[k_x, k_y].$$

The relevant analyses on both the implementation of cross-correlation relationship and the expression of the reference image feature in spatial Fourier domain by designing transfer function E_y have been clarified and added in the main text, as

“In signal processing, cross-correlation is a fundamental operation to measure the similarity of two images or data arrays:

$$f(x) \otimes g(x) = \int dx' f(x') g^*(x' - x) = f(x) \odot g^*(-x) \quad (9)$$

Here, \otimes represents the cross-correlation operation. Correspondingly, by the complex conjugate processing of $E_H(x)$, as $E_H^*(x) \Leftrightarrow \mathcal{F}^*\{E_H\}[-k_x]$, the convolution relationship between the input image and transfer function in Fourier domain in Equation (3) can be transformed into cross-correlation operation

$$E_1(x_1, y_1) \propto E_0(-x_1, -y_1) \odot \mathcal{F}^*\{E_H\}[-k_x, -k_y] = E_0(-x_1, -y_1) \otimes \mathcal{F}\{E_H\}[k_x, k_y] \quad (10)''$$

Since the functionality of cross-correlation operation is to calculate the similarity of two functions in mathematics, cross-correlation function can be utilized to look for the same features between the input image and the reference image. Significantly, the features of reference images are first built in spatial Fourier spectrum by designing transfer function E_H utilizing Huygens' metasurfaces. Then, the complex conjugate of $E_H(x)$ is employed to transfer the convolution equation to cross-correlation relationship. In the first experiment of the cross-correlation operation, the square #1, part of the input signal, is set to be the reference

image, described as the 2D rectangular function $\mathcal{F}\{E_H\}[k_x] = \text{rect}\left(\frac{k_x}{L_x}\right) \text{rect}\left(\frac{k_y}{L_y}\right)$. Hence, to

build the reference image physically, the transfer function on Huygens' metasurface is designed in the form as

$$E_H^*(x', y') = E_H(x', y') \propto \left[L_x \frac{k}{f_2} \text{sinc} \left(L_x \frac{k}{f_2} x' \right) \right] \times \left[L_y \frac{k}{f_2} \text{sinc} \left(L_y \frac{k}{f_2} y' \right) \right]$$

Fig. 5 c in the main text demonstrates the phase profile and the amplitude distribution on 60×80 Huygens' metasurface.

Figure 5. Cross-correlation operation. (c) The phase profile (left) and the amplitude distribution (right) on the 60×80 Huygens' metasurface as the physical block of the reference square #1

b. Experimental results of the cross-correlation operation should be developed further. Could you try such function with more sophisticated image rather than simple line and vertex imaging?

Response: The authors thank the reviewer for the comment. For more sophisticated image experiment, a sequence with several rectangular patterns is pre-designed as the input and reference image as the verification of cross-correlation operation. The relevant analysis is added in the main text, as

“To extend applications of the proposed computational metasurface, the input image is further designed to be one-dimensional sequence which contains two squares with width $w_1=70\text{mm}$. Then, three reference images which contains one, two or three rectangular patterns with the same width (named as ①,②,③) are implemented in spatial Fourier spectrum respectively utilizing Huygens' metasurfaces (see more details of transfer functions in Supplementary section 7). As shown in **Fig. 5e**, the output intensity of electric field indicates the index of geometrical similarity and the location of peaks can identify the regions where input and reference images contain the same pattern features. Hence, our metasurface-based cross-correlators can also perform sequence alignment, offering opportunities for potential signal detection and even DNA sequencing.”

Figure 5. Cross-correlation operation. (a) The input image consisting of the square #1 with $l_{11} = l_{12} = 60$ mm and the square #2 with $l_{21} = l_{22} = 25$ mm at $f_1 = 100$ mm (b) The reference image of the square #1 (c) The phase profile (left) and the amplitude distribution (right) on the 60×80 Huygens' metasurface as the physical block of the reference square #1 (d) Theoretical calculations (left), numerical simulations (middle) and experimental results (right) of the normalized electric field intensity distribution at the output focal length $f_2 = 100$ mm and 10 GHz working frequency for pattern recognition of the square #1. (e) The cross-correlation operation between the input two-square sequence and the reference image containing one, two or three rectangular patterns with $w_1 = 70$ mm (named as ①, ②, ③ respectively). Besides, The normalized intensity of output electric field along x -axis are demonstrated at focal length $f_1 = f_2 = 600$ mm and 10 GHz working frequency for sequence alignment.

4. Technology review

a. The authors provided technology review in figure 1d in terms of thickness and numerical aperture of the device. However, in this paper there is no clear statement or analysis on numerical aperture. Please provide appropriate additional simulation and experimental results in this regard.

Response: The authors thank the reviewer for the comment. Figure 1d illustrates the thickness and numerical aperture of the Huygens' metasurface for the first-order derivative operation. The clear discussions have been added in the main text, as

“Due to the direct modulation in spatial Fourier spectrum by the Huygens' metasurface, the proposed processor can be optimized by increasing the number of meta-atoms for superior light-gathering ability and resolution. Hence, to detect the details of input rectangular image at 100 mm, the number of meta-atoms is set to be 100×140 to keep NA as high as 0.9. Due to the single-layer structure, the overall thickness of operator is $\sim \lambda/6$, which ensures ultra-compactness and integration of our devices.”

Reviewer#2: An analog computer is a device that uses continuous variations in a given physical phenomenon to perform some certain calculations or processing tasks. The digital computer based on electron flows in microelectronic circuits relies on analog-to-digital conversions, which generally suffers from high energy consumption and low operation speed, making it challenging for massive data processing. In recent years, there are considerable interest in analogue computing in the context of metamaterials research, since the subwavelength structures could implement computing functionalities by leveraging light propagation in suitably engineered artificial photonic materials. This solution enables ultrafast speeds, low loss, subwavelength, and parallel operations, holding the promise to overcome the aforementioned challenges. The manuscript by Wang et al proposed a Fourier-based transfer functions for analog computing upon single-layer Huygens' metasurface. Some basic mathematical operations are performed by modulating complex wavefronts in spatial Fourier domain. There are several issues in the paper.

Response: The authors are grateful for the reviewer's positive feedback and recommendation, by which we are mostly encouraged. We also thank the reviewer for the constructive comments below. We make significant revisions based on the reviewer's suggestions, which greatly helped to increase the content and impact of this work. With this, we hope our revised work would satisfy the stringent standard in publication of *Nature Communications*. In the following, please find our point-to-point specific responses.

1. The title of the paper should be specified, since it looks more like a title of review paper. In particular, it seems to me that the similar topic has been extensively reported by some published papers (see references 20-22 and 32-34). Moreover, some results in the present manuscript have also been partially reported elsewhere. For example, the edge detection with vortex phase has been reported in previous published paper (see reference 21), although the Fourier-based transfer functions have not been mentioned in that paper.

Response: The authors are grateful for the comments. Firstly, to specific our work, the title has been revised as “**Single-layer spatial analog meta-processor for imaging processing**”.

Secondly, the novelty of this paper, as now significantly reinforced according to Reviewer's suggestions, is analyzed in Text S1 of SI , as

“ Text S1 The benchmark table of metastructure-based analog processors

To address the novelty and significance, the benchmarks of our work in contrast to other categories of up-to-date metastructure-based analog processors are listed in terms of working mechanism, device design, device dimension, computing throughput, function diversity, as listed in Table S1. Particularly, according to the digital throughput measured in bits per second (bit/s) , the throughput of analog processor here is evaluated by the processing range of input data per unit of time.

Firstly, both the GF kernel and our proposed processor reveal a distinct advantage in device integration for the single-layer spectrum modulation within sub-wavelength scale. Secondly, compared with the increase of waveguide number with more complicated inverse design of metastructure and the device optimization of GF kernel for wider modulation range of incident angle, the computing throughput can boost enormously by simply increasing the metaatom number of $4f$ Fourier system and our proposed metasurface. Finally, compared with other three methods, the angle-dependent response of GF kernel is suitable for limited Fourier-domain operations (such as derivative and integral). For instance, the differentiation is

transformed into the easier multiplication in angular-scattering spectrum, as $E_H(k_x) \propto jk_x$, which can be implemented by the mechanism of Fano resonance or surface plasmon polariton (SPP). However, for the mathematical function with drastic fluctuations, such as the cross-correlation in our work as $E_H(k_x, k_y) \propto \left[L_x \frac{k}{f_2} \text{sinc}(L_x k_x) \right] \times \left[L_y \frac{k}{f_2} \text{sinc}(L_y k_y) \right]$, the difficulties in angle-dependent metaatom design will be greatly increased. Overall, our proposed meta-processor renders a high level of aggregation with respect to the integration, throughput, function diversity..”

Table S1. Benchmarks of the recently proposed metastructure-based analog processors. The cases highlighted in pale green color are the best implementable computing performances with specific methods compared with others.

	The inverse-designed computational metastructure	4f Fourier system	Green's Function (GF) kernel	Our work
Working mechanism	Spatial-domain calculation	Spatial-frequency-domain calculation	Angular-scattering spectrum calculation	Spatial-frequency-domain calculation
Device design	Inverse-designed metamaterial structure	Two graded refractive index (GRIN) lenses and metasurface	Angle-dependent device	Single-layer metasurface
Device dimension/ thickness	Several wavelength scale	Three layers	Single layer	Single layer
Function diversity	Versatile spatial-domain function types	Versatile Fourier-domain function types	Basic Fourier-domain operations	Versatile Fourier-domain function types

Here is the explicit explanation of function diversity. GF method is to design the angle-dependent response of optical metaatoms. The differentiation or integration is transformed into the easier multiplication in spatial Fourier spectrum, as $E_H(x') \propto jx' \Leftrightarrow \mathcal{F}\{E_H\}[k_x] \propto \frac{d}{dk_x} \delta(k_x)$. However, for complex mathematical operation, such as the cross-correlation with image with specific rectangular features proposed in our work, like

$$E_H^*(x', y') = E_H(x', y') \propto \left[L_x \frac{k}{f_2} \text{sinc}\left(L_x \frac{k}{f_2} x'\right) \right] \times \left[L_y \frac{k}{f_2} \text{sinc}\left(L_y \frac{k}{f_2} y'\right) \right],$$

the difficulty in angle-dependent metaatom design will be greatly increased and influence the function diversity.

Moreover, as the reviewer commented, the electric field distribution of our proposed

metasurface-based processor for 2D-edge detection seems like the spiral phase plate in reference 21 of the main text. However, as shown in the Fig. R2.1, the phase variations between two works are actually different. And the amplitude in our work varies with the aperture radius while it keeps uniform in reference 21. In terms of working mechanism, reference 21 is $4f$ Fourier system where the metasurface works as a spatial filter in Fourier spectrum to obtain the edge enhanced image. Our proposed processor work is a $2f$ -structure and the functionalities of our proposed metasurface contains the construction of convolution relationship with the input image and spatial-Fourier-spectrum modulation. Hence, our work can compress the typical $4f$ optical system into the $2f$ structure.

Figure R2.1. The comparison of aperture function for edge detection. (a) The specific phase (upper) and amplitude (lower) profiles on our proposed Huygens' metasurface. (b) The ideal spiral phase with uniform amplitude implemented in the metasurface from reference [21].

Overall, we have significantly reinforced the Introduction section to further address the novelty of the work, as: “By introducing specific phase factor into the complex wavefront profile on the interface, the single-layer meta-structure herein directly tailors the transverse wavevector exerted in spatial Fourier spectrum and hence imposes the customized transfer function for analog image processing, which can essentially compress the typical $4f$ optical system to the $2f$ structure”. Moreover, the advantages has been highlighted and added in the text, as “Our work could enable the real-time and high-throughput parallel computing tasks, overcoming the existing integration issues of traditional bulky Fourier optical devices while performing diverse mathematical operations in contrast to GF kernels. Hence, our proposed miniaturized meta-processor reveals high-performance computing and can be readily generalized for tremendous tasks in analog imaging processing [35,36] and computations such as equation solvers [19,37], edge detection of patterns [26,34], optical memory [38], machine learning [39-41] and others.”

2. In the first paragraph of page 6, the authors write “Fig. 4b, d display the theoretical, numerical and measured electric-field intensity profiles, which show an excellent agreement. One could clearly see strong signals at the edge location of images, verifying the 1D-edge detection capabilities of the proposed analog image processors based on Huygens’ metasurfaces.” In surprise, it is almost impossible to see the edge enhanced image from the

figures (especially in simulations and experimental results). Therefore, I think their claim is, to some extent, exaggerated and misleading.

Response: The authors thank the reviewer for the comment. There are indeed some deviations between the theory, simulations and measurements in Fig. 4b and d mainly due to the limited number of metaatoms. To develop the proposed processor, we have done attempts to improve the numerical experimental results of the first-order derivative operation on input rectangle. Besides, for further optimization of our work, we have added new validations of both the edge detection and cross-correlation operation on more sophisticated images.

Firstly, we have made significant efforts to further optimize our experiments. Experimental results of x-axis edge detection, y-axis edge detection, vertex detection and 2D edge detection on rectangular image have been optimized by increasing the number of meta-atoms from 50×90 to 100×140 and NA from 0.78 to 0.9. As shown in Fig. R22, the edge width and background noise are significantly decreased when a larger metasurface is used.

Figure R2.2. The comparison diagram of normalized electric field distribution for x-axis edge detection, y-axis edge detection, vertex detection and 2D edge detection utilizing 50×90 (left) and 100×140 (right) Huygens' metasurface.

Furthermore, more sophisticated image patterns (such as regular pentagon and hexagon) are processed by 120×170 Huygens' metasurface for 2D-edge detection as the proof-of-concept experiments. The relevant discussions have been added in the main text, as

“Moreover, to validate the maneuverability of our proposed processor across versatile applications, the 2D-edge detection of more complex patterns, including regular pentagon and hexagon, are also implemented by increasing meta-atom number to 120×170 . As illuminated in Fig. 4i, j, the 2D first-order derivative operations are completely performed on both images for edge information extraction.”

Figure 4. The first-order derivative operation. (a) Schematic diagram and (b) Theoretical calculations (left), numerical simulations (middle) and experimental results (right) of the normalized electric field intensity distribution at the output focal length $f_2 = 100$ mm and 10 GHz working frequency for 1D-edge detection along x -axis. (c) Schematic diagram and (d) Theoretical calculations (left), numerical simulations (middle) and experimental results (right) of the normalized electric field intensity distribution at the output focal length $f_2 = 100$ mm and 10 GHz working frequency for 1D-edge detection along y -axis. (e) Schematic diagram and (f) Theoretical calculations (left), numerical simulations (middle) and experimental results (right) of normalized electric field intensity distribution at the output focal length $f_2 = 100$ mm and 10 GHz working frequency for vertex detection. (g) Schematic diagram and (h) Theoretical calculations (left), numerical simulations (middle) and experimental results (right) of normalized electric field intensity distribution at the output focal length $f_2 = 100$ mm and 10 GHz working frequency for 2D-edge detection. (i) The input image of regular pentagon (left) with $l_p = 210$ mm and output experimental results (right) of normalized electric field intensity distribution at the output focal length $f_2 = 100$ mm and 10 GHz working frequency for 2D-edge detection. (j) The input image of regular hexagon (left) with $l_h = 175$ mm and the output experimental results (right) of normalized electric field intensity distribution at the output focal length $f_2 = 100$ mm and 10 GHz working frequency for 2D-edge detection.

Secondly, for more sophisticated image experiment of cross-correlation operation, a sequence with several rectangular patterns is pre-designed as the input and reference image as the verification of cross-correlation operation. The relevant analysis is added in the main text, as

“To extend applications of the proposed computational metasurface, the input image is further designed to be one-dimensional sequence which contains two squares with width $w_1=70\text{mm}$. Then, three reference images which contains one, two or three rectangular patterns with the same width (named as ①,②,③) are implemented in spatial Fourier spectrum respectively utilizing Huygens' metasurfaces (see more details of transfer functions in Supplementary section 7). As shown in **Fig. 5e**, the output intensity of electric field indicates the index of geometrical similarity and the location of peaks can identify the regions where input and reference images contain the same pattern features. Hence, our metasurface-based cross-correlators can also perform sequence alignment, offering opportunities for potential signal detection and even DNA sequencing.”

Figure 5. Cross-correlation operation. (a) The input image consisting of the square #1 with $l_{11} = l_{12} = 60$ mm and the square #2 with $l_{21} = l_{22} = 25$ mm at $f_1 = 100$ mm (b) The reference image of the square #1 (c) The phase profile (left) and the amplitude distribution (right) on the 60×80 Huygens' metasurface as the physical block of the reference square #1 (d) Theoretical calculations (left), numerical simulations (middle) and experimental results (right) of the normalized electric field intensity distribution at the output focal length $f_2 = 100$ mm and 10 GHz working frequency for pattern recognition of the square #1. (e) The cross-correlation operation between the input two-square sequence and the reference image containing one, two or three rectangular patterns with $w_1 = 70$ mm (named as ①, ②, ③ respectively). Besides, The normalized intensity of output electric field along x-axis are demonstrated at focal length $f_1 = f_2 = 600$ mm and 10 GHz working frequency for sequence alignment.

Besides, the derivation of phase profiles on Huygens' metasurface is added in Text S7 of

Supplementary Information for cross-correlation on one-dimensional sequence:

“Text S7 The wavefront profiles on Huygens’ metasurface for cross-correlation on one-dimensional sequence

For the validation of the proposed cross-operator, three reference sequences ①, ② and ③ are built by Huygens' metasurfaces respectively to test the similarity with the input two-square-pulse image. With the same method as the first cross-correlation experiment, the features of three reference sequences are built by designing the transfer function first, as

$$E_H(x', y') \propto \left[L_x \frac{k}{f_2} \text{sinc} \left(L_x \frac{k}{f_2} x' \right) \right] \text{ for } \textcircled{1}$$

$$E_H(x', y') \propto \left[L_x \frac{k}{f_2} \text{sinc} \left(L_x \frac{k}{f_2} x' \right) \right] \times \exp \left(\frac{ik}{f_2} s_1 x' \right) \text{ for } \textcircled{2}$$

$$E_H(x', y') \propto \left[L_x \frac{k}{f_2} \text{sinc} \left(L_x \frac{k}{f_2} x' \right) \right] \times \left[\exp \left(\frac{ik}{f_2} s_1 x' \right) + \exp \left(\frac{ik}{f_2} s_2 x' \right) \right] \text{ for } \textcircled{3}$$

Figure S6. Schematic of three reference sequences ①, ② and ③ with one, two or three rectangular pulses respectively.

Then, the complex conjugate operation of $E_H(x)$ is performed to transfer the convolution equation to cross-correlation relationship for sequence alignment. As shown in Figure S5, according to Equation (2), the aperture function on Huygens' metasurface can be obtained by multiplying the complex conjugate transfer function $E_H^*(x', y')$ with the phase

factor $\exp \left[-\frac{ik}{2f} (x'^2 + y'^2) \right]$, as

$$E_{meta}(x', y') \propto \exp \left[-\frac{ik}{2f} (x'^2 + y'^2) \right] \times E_H^*(x', y') \text{ for } \textcircled{1}, \textcircled{2} \text{ and } \textcircled{3}.”$$

Figure S7. Cross-correlation operation. (a) The phase profile and (b) The amplitude distribution on the 140×30 Huygens' metasurface for the cross-correlation operation on the input two-square sequence and ① sequence. (c) The phase profile and (d) The amplitude distribution on the 140×30 Huygens' metasurface for the cross-correlation operation on the input two-square sequence and ② sequence. (e) The phase profile and (f) The amplitude distribution on the 140×30 Huygens' metasurface for the cross-correlation operation on the input two-square sequence and ③ sequence.

3. In the proposed scheme, the authors attempt to realize the transfer function by manipulating the wavefront with a single-layer Huygens' metasurface. It is known that the edge detection can be evaluated by the spectrum transfer function. However, the absence of spectrum transfer function casts a doubt on the validity of claimed results of the paper. All of these make the whole protocol doubtful.

Response: As the reviewer commented, the complex aperture profiles of single-layer structure design should be analyzed judiciously to verify the theoretical feasibility. Significantly, the introduction of phase factor will contribute to the direct modulation of spatial Fourier spectrum of the input image by transfer function E_H on Huygens' metasurface. For clarity and conciseness, the derivation procedure of the aperture function on Huygens' metasurface, which includes the phase factor and the transfer function, is analyzed and added in the Text S2 section of Supplementary Information, as

“Text S2 Derivation of the aperture function on Huygens' metasurface processor

Under paraxial approximation and in Fresnel regime, the wave in output plane (dubbed as E_1), regarding the impinging image (E_0) through the Huygens' metasurface (E_{meta}), can be expressed as

$$\begin{aligned}
E_1(x_1, y_1) &= -\frac{k^2}{4\pi^2 f_1 f_2} \exp[ik(f_1 + f_2)] \iint_{\Sigma_{meta}} dx'dy' \left\langle \iint_{\Sigma_0} dx_0 dy_0 E_0(x_0, y_0) \exp\left\{\frac{ik}{2f_1}[(x' - x_0)^2 + (y' - y_0)^2]\right\} \right\rangle \\
&\quad E_{meta}(x', y') \exp\left\{\frac{ik}{2f_2}[(x_1 - x')^2 + (y_1 - y')^2]\right\} \\
&= -\frac{k^2}{4\pi^2 f_1 f_2} \exp[ik(f_1 + f_2)] \exp\left(\frac{ik}{2f_2}x_1^2 + \frac{ik}{2f_2}y_1^2\right) \iint_{\Sigma_0} dx_0 dy_0 E_0(x_0, y_0) \exp\left(\frac{ik}{2f_2}x_0^2 + \frac{ik}{2f_2}y_0^2\right) \\
&\quad \iint_{\Sigma_{meta}} dx'dy' E_{meta}(x', y') \exp\left[ik\left(\frac{x_0}{f_1} + \frac{x_1}{f_2}\right)x' + ik\left(\frac{y_0}{f_1} + \frac{y_1}{f_2}\right)y'\right] \exp\left\{\frac{ik}{2}\left(\frac{1}{f_1} + \frac{1}{f_2}\right)[x'^2 + y'^2]\right\} \quad (S1)
\end{aligned}$$

From Equation S1, to construct the Fourier transform function in form of $\exp[ikx' +iky']$, the quadratic term $\exp\left\{\frac{ik}{2}\left(\frac{1}{f_1} + \frac{1}{f_2}\right)[x'^2 + y'^2]\right\}$ should be eliminated by introducing the concave-lens phase factor $\exp\left[-\frac{ik}{2f}(x'^2 + y'^2)\right]$ on the Huygens' metasurface aperture, where $\frac{1}{f} = \frac{1}{f_1} + \frac{1}{f_2}$. Hence, the the aperture function on the Huygens' metasurface is defined as

$$E_{meta}(x', y') = \exp\left[-\frac{ik}{2f}(x'^2 + y'^2)\right] E_H(x', y') \quad (S2)$$

where $\frac{1}{f} = \frac{1}{f_1} + \frac{1}{f_2}$. Then, equation S1 can be derived as:

$$\begin{aligned}
E_1(x_1, y_1) &= -\frac{k^2}{4\pi^2 f_1 f_2} \exp[ik(f_1 + f_2)] \exp\left(\frac{ik}{2f_2}x_1^2 + \frac{ik}{2f_2}y_1^2\right) \iint_{\Sigma_0} dx_0 dy_0 E_0(x_0, y_0) \exp\left(\frac{ik}{2f_2}x_0^2 + \frac{ik}{2f_2}y_0^2\right) \\
&\quad \iint_{\Sigma_{meta}} dx'dy' E_H(x', y') \exp\left[ik\left(\frac{x_0}{f_1} + \frac{x_1}{f_2}\right)x' + ik\left(\frac{y_0}{f_1} + \frac{y_1}{f_2}\right)y'\right] \\
&= -\frac{k^2}{2\pi f_1 f_2} \left\{ \exp[ik(f_1 + f_2)] \exp\left(\frac{ik}{2f_2}x_1^2 + \frac{ik}{2f_2}y_1^2\right) \right\} \iint_{\Sigma_0} dx_0 dy_0 E_0(x_0, y_0) \exp\left(\frac{ik}{2f_2}x_0^2 + \frac{ik}{2f_2}y_0^2\right) \\
&\quad \mathcal{F}\{E_H(x', y')\} \left[\frac{k}{f_1}(x_0 - \tilde{x}_1), \frac{k}{f_1}(y_0 - \tilde{y}_1) \right] \\
&= -\frac{k^2}{2\pi f_1 f_2} \{ \dots \} \left\{ E_0(\tilde{x}_1, \tilde{y}_1) \exp\left(\frac{ik}{2f_2}\tilde{x}_1^2 + \frac{ik}{2f_2}\tilde{y}_1^2\right) \right\} \odot \mathcal{F}\{E_H(x', y')\} \left[-\frac{k}{f_1}\tilde{x}_1, -\frac{k}{f_1}\tilde{y}_1 \right] \quad (S3)
\end{aligned}$$

where $\tilde{x}_1 = -\frac{f_1}{f_2}x_1$, $\tilde{y}_1 = -\frac{f_1}{f_2}y_1$ and \odot represents two-dimensional convolution operation. Via applying the convolution theorem,, described as $\mathcal{F}\{f(x) \odot g(x)\} = \mathcal{F}\{f(x)\}[k_x] \times \mathcal{F}\{g(x)\}[k_x]$, the output image in Fourier spectrum can be obtained by multiplying input signals with repeated Fourier transforms of E_H , as $\mathcal{F}\{\mathcal{F}\{E_H(x', y')\}\} = E_H(-x', -y')$. Hence, as the ratio of the output signal and input signal in Fourier spectrum, E_H acts as transfer function and x' and y' indicate the spatial Fourier frequencies (wavevectors). Huygens' metasurface can directly modulate the spatial Fourier spectrum through E_H for predesigned analog processing. For direct expression of formula in

the main text, Equation (S3) can be described as:

$$E_1(x_1, y_1) = \left\{ \frac{k^2}{2\pi f_1 f_2} \exp[ik(f_1 + f_2)] \exp\left[\frac{ik}{2f_2}(x_1^2 + y_1^2)\right] \right\} \left\{ E_0\left(-\frac{f_1}{f_2}x_1, -\frac{f_1}{f_2}y_1\right) \phi(x_1, y_1) \right\} \odot \mathcal{F}\{E_H(x', y')\}[k_x, k_y] \quad (\text{S4})$$

where the additional phase factor $\phi(x_1, y_1) = \exp\left[\frac{ikf_1}{2f_2^2}(x_1^2 + y_1^2)\right]$ and

$k_x = \frac{k}{f_2}x_1$ and $k_y = \frac{k}{f_2}y_1$. Overall, by superimposing the specific phase factor related with the

input and output focal length $\exp\left[-\frac{ik}{2f}(x'^2 + y'^2)\right]$ on the transfer function (E_H)

algorithmically, the proposed Huygens' metasurface can directly manipulate spatial Fourier frequencies for the target outputs with single-layer structure."

Equation (S4) in Supplementary Information or Equation (3) in the main text demonstrate the convolution relationship between the input input image and the spatial-frequency-domain $E_H(x', y')$ on the Huygens' metasurfaces. Then, the transfer function is introduced to describe the modulation in spatial Fourier spectrum utilizing Huygens' metasurface. The definition, functionality and examples of transfer function are added in the main text, as

"Via applying the convolution theorem $\mathcal{F}\{f(x) \odot g(x)\} = \mathcal{F}\{f(x)\} \times \mathcal{F}\{g(x)\}$ to Equation (3), the output image in Fourier spectrum can be obtained by multiplying input signals with repeated Fourier transforms of E_H , as

$$\begin{aligned} \mathcal{F}\{E_1(x_1, y_1)\} &= \{\dots\} \mathcal{F}\{E_0(x_1, y_1) \phi(x_1, y_1)\} \times \mathcal{F}\{\mathcal{F}\{E_H(x', y')\}\} \\ &= \{\dots\} \mathcal{F}\{E_0(x_1, y_1) \phi(x_1, y_1)\} \times E_H(-x', -y') \end{aligned} \quad (3)$$

Since the definition of transfer function is the ratio of the output signal to input signal in

Fourier spectrum, as $E_H(-x', -y') = \frac{\mathcal{F}\{E_1(x_1, y_1)\}}{\mathcal{F}\{E_0(x_1, y_1) \phi(x_1, y_1)\}}$, E_H can model the

spatial-frequency response of the proposed processor quantitatively and map the output image for each possible input. For maneuverability without loss of generality, the transfer function of first-order differential operator is $E_H(x) \propto jx$ and for the integrator, transfer function can be

described as $E_H(x) \propto \frac{1}{jx}$."

Overall, we can superimpose the specific phase factor related with the input and output focal

length $\exp\left[-\frac{ik}{2f}(x'^2 + y'^2)\right]$ on the transfer function (E_H) algorithmically. In this manner,

the proposed single-layer Huygens' metasurface can directly implement spatial frequencies for the target output signal, which avoids auxiliary optical elements.

Reviewer #3: The authors reported a revised analog processor based on the spatial Fourier filtering method using metasurfaces. Compared to traditional $4f$ system, a single-layer metasurface is used to achieve the function in a more compact $2f$ system. Comparing to Green's function method, it has more design freedom and flexibility. Optical analog computing using meta-optics is an interesting and promising research field. The work made a good contribution to the development of this field and showed a feasible way of reducing the size of the meta-optics system. Below are some points should be addressed by the authors:

Response: The authors are grateful for the reviewer's endorsement and positive comments. We do appreciate that the reviewer shared the same excitement as we do. We also thank the reviewer for the comments in the following, which have helped us to improve the work a lot. Below, please find our specific point-to-point responses.

1) The novelty and merits of the work need to be justified clearly. One key novelty, as the authors stated, "Hence, we propose a single-layer metasurface-based analog processor that essentially compresses the typical $4f$ optical system into the $2f$ structure." However, those reference works listed by the authors, like Zhou 2020 and Cordano 20219, are really "single-layer" devices which in principle impose no restraints upon the object and image positions, while this work is actually a " $2f$ " device of which the input plane and output plane should be put at the positions of $-f$ and f . In addition, the authors picked the thickness and NA as the two figure of merits to benchmark with other works in Figure 1D. For metasurfaces like that of Zhou 2020 and Cordano 20219, both thickness and NA can be easily designed in the reasonable ranges. Figure 1D doesn't help to justify the novelty and impact of the work.

Response: The authors thank the reviewer for the comment. Firstly, the specific point about the requirement of the fixed focal length in our method have been clarified and added in the main text. Zhou 2020 and Cordano 2019 have demonstrate the direct modulation of transfer functions in momentum space by utilizing Fano resonances for first- and second-order spatial differentiation. These two methods, which essentially belong to Green's Function (GF) kernel, can directly operate on the angular scattering spectrum of impinging waves, dramatically decreases the overall space of processor. As the reviewer commented, methods of Zhou 2020 and Cordano 2019 impose no restraints upon the object and image positions. By contrast, our work needs to keep input image at the fixed position of $-f_1$ and the output plane at focal length f_2 . Hence, the restriction of our work is pointed out in the Introduction section of the main text, as

"Hence, we propose a single-layer metasurface-based analog processor (**Fig. 1c**). Such proposal **with restrictions on the fixed input and output focal length** overcomes the existing issues of compactness and maneuverability of traditional bulky Fourier optical devices and shows high flexibility compared to GF approach."

Secondly, the key novelty and merits of this paper, including the device integration, function flexibility and computing throughput, as reinforced according to Reviewer's suggestions, is analyzed in Text S1 of SI, as

"Text S1 The benchmark table of metastructure-based analog processors

To address the novelty and significance, the benchmarks of our work in contrast to other categories of up-to-date metastructure-based analog processors are listed in terms of working mechanism, device design, device dimension, computing throughput, function diversity, as listed in Table S1. Particularly, according to the digital throughput measured in bits per second

(bit/s) , the throughput of analog processor here is evaluated by the processing range of input data per unit of time.

Firstly, both the GF kernel and our proposed processor reveal a distinct advantage in device integration for the single-layer spectrum modulation within sub-wavelength scale. Secondly, compared with the increase of waveguide number with more complicated inverse design of metastructure and the device optimization of GF kernel for wider modulation range of incident angle, the computing throughput can boost enormously by simply increasing the metaatom number of $4f$ Fourier system and our proposed metasurface. Finally, compared with other three methods, the angle-dependent response of GF kernel is suitable for limited Fourier-domain operations (such as derivative and integral). For instance, the differentiation is transformed into the easier multiplication in angular-scattering spectrum, as $E_H(k_x) \propto jk_x$, which can be implemented by the mechanism of Fano resonance or surface plasmon polariton (SPP). However, for the mathematical function with drastic fluctuations, such as the cross-correlation in our work as $E_H(k_x, k_y) \propto \left[L_x \frac{k}{f_2} \text{sinc}(L_x k_x) \right] \times \left[L_y \frac{k}{f_2} \text{sinc}(L_y k_y) \right]$, the difficulties in angle-dependent metaatom design will be greatly increased. Overall, our proposed meta-processor renders a high level of aggregation with respect to the integration, throughput, function diversity. ”

Table S1. Benchmarks of the recently proposed metastructure-based analog processors. The cases highlighted in pale green color are the best implementable computing performances with specific methods compared with others.

	The inverse-designed computational metastructure	$4f$ Fourier system	Green's Function (GF) kernel	Our work
Working mechanism	Spatial-domain calculation	Spatial-frequency-domain calculation	Angular-scattering spectrum calculation	Spatial-frequency-domain calculation
Device design	Inverse-designed metamaterial structure	Two graded refractive index (GRIN) lenses and metasurface	Angle-dependent device	Single-layer metasurface
Device dimension/ thickness	Several wavelength scale	Three layers	Single layer	Single layer
Function diversity	Versatile spatial-domain function types	Versatile Fourier-domain function types	Basic Fourier-domain operations	Versatile Fourier-domain function types

Here is the explicit explanation of function diversity. GF method is to design the angle-dependent response of optical metaatoms. The differentiation or integration is transformed into the easier multiplication in spatial Fourier spectrum, as $E_H(x') \propto jx' \Leftrightarrow \mathcal{F}\{E_H\}[k_x] \propto \frac{d}{dk_x} \delta(k_x)$. However, for complex mathematical operation,

such as the cross-correlation with image with specific rectangular features proposed in our work, like

$$E_H^*(x', y') = E_H(x', y') \propto \left[L_x \frac{k}{f_2} \text{sinc}\left(L_x \frac{k}{f_2} x'\right) \right] \times \left[L_y \frac{k}{f_2} \text{sinc}\left(L_y \frac{k}{f_2} y'\right) \right],$$

the difficulty in angle-dependent metaatom design will be greatly increased and influence the function diversity.

Thirdly, the elaborate discussion on the thickness and NA have been added in the main text, as

“Due to the direct modulation in spatial Fourier spectrum by the Huygens’ metasurface, the proposed processor can be optimized by increasing the number of meta-atoms for superior light-gathering ability and resolution. Hence, to detect the details of input rectangular image at 100 mm, the number of meta-atoms is set to be 100 ×140 to keep NA as high as 0.9. Due to the single-layer structure, the overall thickness of operator is $\sim\lambda/6$, which ensures ultra-compactness and integration of our devices.”

Finally, to further address the novelty and impact of the work, as: “By introducing specific phase factor into the complex wavefront profile on the interface, the single-layer meta-structure herein directly tailors the transverse wavevector exerted in spatial Fourier spectrum and hence imposes the customized transfer function for analog image processing, which can essentially compress the typical $4f$ optical system to the $2f$ structure.” Moreover, the advantages has been highlighted and added in the text, as “Our work could enable the real-time and high-throughput parallel computing tasks, overcoming the existing integration issues of traditional bulky Fourier optical devices while performing diverse mathematical operations in contrast to GF kernels. Hence, our proposed miniaturized meta-processor reveals high-performance computing and can be readily generalized for tremendous tasks in analog imaging processing [35,36] and computations such as equation solvers [19,37], edge detection of patterns [26,34], optical memory [38], machine learning [39-41] and others.”

2) The designs are demonstrated in the microwave frequency, which are usually expected to have more satisfactory results in both simulation and experiments than in optical frequency considering the better fabrication accuracy, less materials limitation and even more design freedom. However, the results showed in Figure 4 are far from satisfactory, even not comparing with other reports but comparing with its own theoretical results. For example, the simulation results in Figure 4 b and d are all broken spots instead of lines showed in Figure 4 a and c. Actually, the simulated result in Figure 4d resembles more of the theoretical result for vortex detection in Figure 4e instead of that for 1D edge detection along y-axis in Figure 4c. These project poor quality to the work.

Response: As the reviewer commented, considering the fabrication accuracy, optional materials category and design freedom, metasurfaces in microwave frequency actually reveal distinct advantages over optical metastructures. Hence, microwave metasurface enables the higher device transmittance efficiency, more multiplexing channels, better manipulation ability of complex wavefront and so on. However, the image resolution and overall device dimension are quite limited in terms of the overall size normalized by the (millimeter) working wavelength.

Nevertheless, we indeed performed more simulations and experiments to further

improved the experimental results. To develop the proposed processor, we attempted to improve the experimental results of the first-order derivative operation on input rectangle and add new validation of the same function on more sophisticated images.

Firstly, we have made significant efforts to further optimize our experiments. Experimental results of x-axis edge detection, y-axis edge detection, vertex detection and 2D edge detection on rectangular image have been optimized by increasing the number of meta-atoms from 50×90 to 100×140 and NA from 0.78 to 0.9. As shown in Fig. R3.1, the edge width and background noise are significantly decreased when a larger metasurface is used.

Figure R3.1. The comparison diagram of normalized electric field distribution for x-axis edge detection, y-axis edge detection, vertex detection and 2D edge detection utilizing 50×90 (left) and 100×140 (right) Huygens' metasurface.

Furthermore, more sophisticated image patterns (such as regular pentagon and hexagon) are processed by 120×170 Huygens' metasurface for 2D-edge detection as the proof-of-concept experiments. The relevant discussions have been added in the main text, as

“Moreover, to validate the maneuverability of our proposed processor across versatile applications, the 2D-edge detection of more complex patterns, including regular pentagon and hexagon, are also implemented by increasing meta-atom number to 120×170 . As illuminated in Fig. 4i, j, the 2D first-order derivative operations are completely performed on both images for edge information extraction.”

Figure 4. The first-order derivative operation. (a) Schematic diagram and (b) Theoretical calculations (left), numerical simulations (middle) and experimental results (right) of the normalized electric field intensity distribution at the output focal length $f_2 = 100$ mm and 10 GHz working frequency for 1D-edge detection along x -axis. (c) Schematic diagram and (d) Theoretical calculations (left), numerical simulations (middle) and experimental results (right) of the normalized electric field intensity distribution at the output focal length $f_2 = 100$ mm and 10 GHz working frequency for 1D-edge detection along y -axis. (e) Schematic diagram and (f) Theoretical calculations (left), numerical simulations (middle) and experimental results (right) of normalized electric field intensity distribution at the output focal length $f_2 = 100$ mm and 10 GHz working frequency for vertex detection. (g) Schematic diagram and (h) Theoretical calculations (left), numerical simulations (middle) and experimental results (right) of normalized electric field intensity distribution at the output focal length $f_2 = 100$ mm and 10 GHz working frequency for 2D-edge detection. (i) The input image consisting of regular pentagon (left) with $l_p = 210$ mm and the output experimental results (right) of normalized electric field intensity distribution at the output focal length $f_2 = 100$ mm and 10 GHz working frequency for 2D-edge detection. (j) The input image consisting of regular hexagon (left) with $l_h = 175$ mm and the output experimental results (right) of normalized electric field intensity distribution at the output focal length $f_2 = 100$ mm and 10 GHz working frequency

for 2D-edge detection. (i) The input image of regular pentagon (left) with $l_p=210$ mm and output experimental results (right) of normalized electric field intensity distribution at the output focal length $f_2=100$ mm and 10 GHz working frequency for 2D-edge detection. (j) The input image of regular hexagon (left) with $l_h=175$ mm and the output experimental results (right) of normalized electric field intensity distribution at the output focal length $f_2=100$ mm and 10 GHz working frequency for 2D-edge detection.

3. Even though NA is used as one of the two merits for benchmarking, it is not discussed in the main text except appearing in Figure 1d. The authors shall give some elaborations of the importance and difficulties of achieving such a NA and the calculation of the NA in this work.

Response: The authors thank the reviewer for the comment. Figure 1d illustrates the thickness and numerical aperture of the Huygens' metasurface for the first-order derivative operation. The clear discussions on the implementation, functionality and the calculation parameters of NA have been added in the main text, as

“Due to the direct modulation in spatial Fourier spectrum by the Huygens’ metasurface, the proposed processor can be optimized by increasing the number of meta-atoms for superior light-gathering ability and resolution. Hence, to detect the details of input rectangular image at 100 mm, the number of meta-atoms is set to be 100×140 to keep NA as high as 0.9. Due to the single-layer structure, the overall thickness of operator is $\sim \lambda/6$, which ensures ultra-compactness and integration of our devices.”

4. Please indicate the resolution of this device, which is an important indicator of the analog processor.

Response: The authors thank the reviewer for the comment. The discussions on the resolution of both the first-order derivative and cross-correlation operation are added in the in Text S6 of Supplementary Information, as

“Text S6 The resolution of the first-order derivative and cross-correlation operation

As an indicator of the proposed analog processor, the system resolution of edge detection and cross-correlation operation is analyzed respectively. As shown in Fig. S6 a and b, the 2D edges of rectangles with different sizes are clearly demonstrated along both horizontal and vertical directions. When the minimal edge length decreases to 20mm, the detected edge reveals intermittent and a dramatic increase of sidelobe intensity influences the edge extraction. Hence, the resolution of the proposed first-order derivative operator resolution can be considered to be roughly 25mm.

Figure S4. Output electric field intensity of 2D-edge detection on rectangles with different sizes of (a) $l_1=36$ mm, $l_2=30$ mm, (b) $l_1=30$ mm, $l_2=25$ mm, (c) $l_1=25$ mm, $l_2=20$ mm and (d) $l_1=25$ mm, $l_2=20$ mm.

Moreover, the cross-correlation operation between the input with two squares and reference image with one rectangular pattern is analyzed. As shown in Fig. S7, the two rectangular pulse of width can not be distinguished until the interval w_2 reaches to the resolution of 40mm.”

Figure S5. Output electric field intensity of the cross-correlation between the one-rectangular-pattern sequence and the input two-square sequence with width $w_1=70\text{mm}$ and interval (a) $w_2=20\text{mm}$, (b) $w_2=38\text{mm}$, (c) $w_2=40\text{mm}$, (d) $w_2=50\text{mm}$, (e) $w_2=70\text{mm}$, (f) $w_2=90\text{mm}$.

5. Figure 1a and b look very nice. Why the authors didn't use them such as the institution's logo to demonstrate the edge detection and pattern recognition? That would make the illustration and real work more consistent and more attractive. Are there any practical challenges to apply the methodology described in the work to a more complex object than e.g. the simple square showed in Figure 5?

Response: The authors thank the reviewer for this comment. Since the working wavelength is in millimeter range, the overall dimension of metasurface and resolution are quite large in contrast to optical detection, which makes the proposed analog processor probably not suitable for the edge detection and pattern recognition of such complex logo as Figure 1a and b. On the other hand, we have done attempts to improve experimental results of edge detection on the input rectangle. Furthermore, we have added new validations of the first-order derivative and cross-correlation operation on more sophisticated images for the validity of our methodology.

The improved and newly added edge-detection experiments have been discussed in response 2. Moreover, for experiments of more sophisticated images, a sequence with several rectangular shapes is predesigned as the input and reference image as the verification of cross-correlation operation. The relevant analysis is added in the main text, as

“To extend applications of the proposed computational metasurface, the input image is further designed to be one-dimensional sequence which contains two squares with width $w_1=70\text{mm}$. Then, three reference images which contains one, two or three rectangular patterns with the same width (named as ①,②,③) are implemented in spatial Fourier spectrum respectively utilizing Huygens' metasurfaces (see more details of transfer functions in Supplementary section 7). As shown in Fig. 5e, the output intensity of electric field indicates the index of geometrical similarity and the location of peaks can identify the regions where input and reference images contain the same pattern features. Hence, our metasurface-based cross-correlators can also perform sequence alignment, offering opportunities for potential signal detection and even DNA sequencing.”

Figure 5. Cross-correlation operation. (a) The input image consisting of the square #1 with $l_{11} = l_{12} = 60$ mm and the square #2 with $l_{21} = l_{22} = 25$ mm at $f_1 = 100$ mm (b) The reference

image of the square #1 (c) The phase profile (left) and the amplitude distribution (right) on the 60×80 Huygens' metasurface as the physical block of the reference square #1 (d) Theoretical calculations (left), numerical simulations (middle) and experimental results (right) of the normalized electric field intensity distribution at the output focal length $f_2 = 100$ mm and 10 GHz working frequency for pattern recognition of the square #1. (e) The cross-correlation operation between the input two-square sequence and the reference image containing one, two or three rectangular patterns with $w_1 = 70$ mm (named as ①,②,③ respectively). Besides, The normalized intensity of output electric field along x-axis are demonstrated at focal length $f_1=f_2=600$ mm and 10 GHz working frequency for sequence alignment.

Besides, the derivation of phase profiles on Huygens' metasurface is added in Text S7 of Supplementary Information for cross-correlation on one-dimensional sequence:

“Text S7 The wavefront profiles on Huygens' metasurface for cross-correlation on one-dimensional sequence

For the validation of proposed cross-operator, three reference sequences ①, ② and ③ are built by Huygens' metasurfaces respectively to test the similarity with the input two-square-pulse image. With the same method as the first cross-correlation experiment, the features of three reference sequences are built by designing the transfer function first, as

$$E_H(x', y') \propto \left[L_x \frac{k}{f_2} \text{sinc} \left(L_x \frac{k}{f_2} x' \right) \right] \text{ for } \textcircled{1}$$

$$E_H(x', y') \propto \left[L_x \frac{k}{f_2} \text{sinc} \left(L_x \frac{k}{f_2} x' \right) \right] \times \exp \left(\frac{ik}{f_2} s_1 x' \right) \text{ for } \textcircled{2}$$

$$E_H(x', y') \propto \left[L_x \frac{k}{f_2} \text{sinc} \left(L_x \frac{k}{f_2} x' \right) \right] \times \left[\exp \left(\frac{ik}{f_2} s_1 x' \right) + \exp \left(\frac{ik}{f_2} s_2 x' \right) \right] \text{ for } \textcircled{3}$$

Figure S6. Schematic of three reference sequences ①, ② and ③ with one, two or three rectangular pulses respectively.

Then, the complex conjugate operation of $E_H(x)$ is performed to transfer the

convolution equation to cross-correlation relationship for sequence alignment. As shown in Figure S3, according to Equation (2), the aperture function on Huygens' metasurface can be obtained by multiplying the complex conjugate transfer function $E_H^*(x', y')$ with the phase

factor $\exp\left[-\frac{ik}{2f}(x'^2 + y'^2)\right]$, as

$$E_{meta}(x', y') \propto \exp\left[-\frac{ik}{2f}(x'^2 + y'^2)\right] \times E_H^*(x', y') \text{ for } \textcircled{1}, \textcircled{2} \text{ and } \textcircled{3}."$$

Figure S7. Cross-correlation operation. (a) The phase profile and (b) The amplitude distribution on the 140×30 Huygens' metasurface for the cross-correlation operation operation on the input two-square sequence and $\textcircled{1}$ sequence. (c) The phase profile and (d) The amplitude distribution on the 140×30 Huygens' metasurface for the cross-correlation operation operation on the input two-square sequence and $\textcircled{2}$ sequence. (e) The phase profile and (f) The amplitude distribution on the 140×30 Huygens' metasurface for the cross-correlation operation operation on the input two-square sequence and $\textcircled{3}$ sequence.

6. In Figure 3c, the non-zero amplitude of the meta-atoms are fluctuating or deviating a lot from the set value in the dotted line, especially for amplitude of 0.67 and 1. What is the reason?

Response: The authors thank the reviewer for the comment. The working mechanism of Huygens' metasurface is the modulation of overlap between the crossed electric and magnetic dipole resonances, to realize the full and independent control of both the transmission phase and amplitude. As shown in the Fig. R3.2., the field amplitudes of electric and magnetic dipoles increase drastically at the resonance frequencies. Besides, the closer to the exact resonant frequency (field amplitude peak), much higher the gradient of field intensity

correspondingly will be, which means the transmission amplitude is more sensitive to the overlaps between the electric and magnetic dipoles. Significantly, for the unity transmission, the peaks of electric and magnetic resonances should be accurately overlapped. However, since the limited fabrication accuracy of our PCB technology is 0.2mm, the dimension deviations of the fabricated Huygens' metaatoms will induce bigger fluctuations when the transmission amplitude becomes higher, especially for amplitude of 0.67 and 1. Relevant analysis is added in the main text, as

“Moreover, the deviations between the simulated transmission coefficient and the predesigned value result from the minimum process tolerance, especially for metaatoms with the transmission amplitude of 0.67 and 1, which are more sensitive to the offsets of overlaps between the magnetic and electric resonance.”

Figure R3.2. (a) Analytical results for the electric-field amplitude of the transmitted light (green) and the contributions of the electric (blue) and magnetic (red) resonances for the case of nonoverlapping resonances. b) Analytical phase spectrum of the transmitted electric field (green) and the electric (blue) and magnetic (red) response. c), d) Corresponding diagrams for the case of spectrally overlapping electric and magnetic resonances: c) Field amplitudes, d) phase spectrum.[R1]

R1. Decker, M., Staude, I., Falkner, M., et al. High efficiency dielectric Huygens' surfaces. *Adv. Opt. Mater.* **3**, 813-820 (2015).

7. Photos or images of the fabricated meta-optics are highly recommended to be shown in the manuscript.

Response: As the reviewer commented, photograph of the fabricated Huygens' metasurface are added in the Text S7 of Supplementary Information, as

“ **Text S8 Photograph of the fabricated Huygens' metasurfaces**

Fig. S8 shows the fabricated Huygens' metasurface for edge detection and sequence alignment respectively. The metasurfaces are composed of parallel dielectric on which are printed the Huygens' metaatoms"

Figure S8. The fabricated Huygens' metasurface. (a) The 100×140 Huygens' metasurface for 2D-edge on rectangle (b) The 140×30 Huygens' metasurface for cross-correlation operation. The inset is a zoomed-in view of one-side fabricated Huygens metasurface.

8. For Eq. 10, please check if the middle part of the equation is correct and introduce $R_{ff}(x-t)$ in the text?

Response: The authors are grateful for the comments. The Equation (11) has been modified from

$$f(x) \otimes g(x) = f(x) \otimes f(x-t) = \int dx' f(x') g^*(x'-t+x) = R_{ff}(x-t) \quad (11)$$

to

$$f(x) \otimes g(x) = f(x) \otimes f(x-t) = \int dx' f(x') f^*(x'-t+x) = R_{ff}(x-t) \quad (11)$$

9.) In Figure 5c, seems only central part of the meta-atoms have non-zero amplitude. If that is true, there is no need to design the phase profile for the meta-atoms outside the central part?

Response: As the reviewer commented, when the transmission amplitude of the meta-atom equals to 0, its phase variation essentially has no impact on the manipulation of transmitted field distribution. Therefore, only one metaatom needs to be designed for zero transmission amplitude.

Changes being made to the manuscript:

Firstly, the diagram of metaatom design in Fig. 3c has been modified and relevant discussion is clarified in the main text, as

“Particularly, when the transmission amplitude of metaatom equals to 0, its phase shifts contribute no variations on the transmitted wavefront. Therefore, only one Huygens' meta-atom is designed for transmission amplitude equivalent to 0 with transmission phase of $-\pi$.”

Figure 3. Design of Huygens' metasurface processor. (c) Simulated transmission amplitude and phase of 25 extracted meta-atoms by modulating the geometrical parameters l_e , h_e and l_m , h_m at 10 GHz.

Moreover, since there is no need to design the phase variation when the transmission amplitude equals to 0, the phase profiles in Fig. 5c, Fig. S2 and Fig. S5 have been modified to be $-\pi$ uniformly corresponding to zero amplitude, as

Figure 5. Cross-correlation operation. (c) The phase profile (left) and the amplitude distribution (right) on the 60×80 Huygens' metasurface as the physical block of the reference square #1.

Figure S2. Derivative operation. (a) The phase profile and (b) The amplitude distribution on the 100×140 Huygens' metasurface for the first-order derivative operation along x-axis. (c) The phase profile and (d) The amplitude distribution on the 100×140 Huygens' metasurface for the first-order derivative operation along y-axis. (e) The phase profile and (f) The amplitude distribution on the 100×140 Huygens' metasurface for 2D edge detection. (g) The phase profile and (h) The amplitude distribution on the 100×140 Huygens' metasurface for vertex detection.

Figure S7. Cross-correlation operation. (a) The phase profile and (b) The amplitude distribution on the 140×30 Huygens' metasurface for the cross-correlation operation on the input two-square sequence and ① sequence. (c) The phase profile and (d) The amplitude distribution on the 140×30 Huygens' metasurface for the cross-correlation operation on the input two-square sequence and ② sequence. (e) The phase profile and (f) The amplitude distribution on the 140×30 Huygens' metasurface for the cross-correlation operation on the input two-square sequence and ③ sequence.

REVIEWERS' COMMENTS

Reviewer #1 (Remarks to the Author):

The authors revised the manuscript reflecting all the queries and comments. The novelty of the work is strengthened well with the revised supplementary information and the revised text, which makes me to convince the work. I recommend this article to be published in Nature Communications.

Reviewer #2 (Remarks to the Author):

In the revised version of the manuscript, the authors have properly replied to my questions raised during the first review process. This not only makes the manuscript more insightful, but also makes the whole experimental procedure easier to follow. I support its publication, but the authors should address the following new issues in this revision.

1. The quality of edge enhanced images is still relatively poor than some previously reported results, although it has been significantly enhanced in the revised version. This casts a little doubt on the validity of proposed scheme of the paper. How to further improve the quality of the edge enhanced images?
2. Only several limited simple cases on the differential operations and edge enhanced images have been reported in the present manuscript. As a spatial analog meta-processor, more functions should be available.

Reviewer #3 (Remarks to the Author):

The authors have made efforts to address all the comments and to improve the design and experimental results. The paper is in good shape now.

Reviewer #1 (Remarks to the Author):

Comments: The authors revised the manuscript reflecting all the queries and comments. The novelty of the work is strengthened well with the revised supplementary information and the revised text, which makes me to convince the work. I recommend this article to be published in Nature Communications.

Response: We thank the referee for these positive feedbacks.

Reviewer #2 (Remarks to the Author):

Comments: In the revised version of the manuscript, the authors have properly replied to my questions raised during the first review process. This not only makes the manuscript more insightful, but also makes the whole experimental procedure easier to follow. I support its publication, but the authors should address the following new issues in this revision.

Response: We thank the referee for the positive remarks.

Comments: The quality of edge enhanced images is still relatively poor than some previously reported results, although it has been significantly enhanced in the revised version. This casts a little doubt on the validity of proposed scheme of the paper. How to further improve the quality of the edge enhanced images?

Response: We thank the referee for the comment about the edge enhanced images. We have added the relevant analysis in Text S9 of Supplementary Information.

“To enhance the practicability and maneuverability of the Huygens metasurface processor, the proposed working mechanism has been applied to the optical-spectrum edge detection to improve the image quality, and the Laplace operation is implemented to increase function diversity, respectively. Firstly, compared with microwave differentiator, the first-order derivative operation performed at optical frequencies can extract micrometer-scale edges and enable detection of sophisticated images.”

Comments: Only several limited simple cases on the differential operations and edge enhanced images have been reported in the present manuscript. As a spatial analog meta-processor, more functions should be available.

Response: We thank the referee for the comment. To further develop the proposed processor for versatile functions, the Laplace operator, of an essential role in the fields of image processing and biomedical science, could also be realized as an auxiliary example, which we analyzed in Text S9 of Supplementary Information, as:

“Then, the Laplace operator, as the isotropic second-order differentiation ($\nabla^2 E_0(x_1, y_1) = \left| \frac{d^2 E_0(x_1, y_1)}{dx_1^2} + \frac{d^2 E_0(x_1, y_1)}{dy_1^2} \right|$, where ∇^2 indicates the Laplace operator and $E_0(x_1, y_1)$

denotes the input electric field), can be performed by our proposed metaprocessor. According to Equation (2) and (4), by designing the transfer function $E_H(x', y')$ as

$$E_H(x', y') \propto -x'^2 - y'^2, \quad (S9)$$

Huygens metasurface can extract the edge details of the 2D object.”

Reviewer #3 (Remarks to the Author):

Comments: The authors have made efforts to address all the comments and to improve the design and experimental results. The paper is in good shape now.

Response: We are grateful for the referee’s positive feedback and recommendation.